# Meta Learning Backpropagation And Improving It

**Louis Kirsch**[1], **Jürgen Schmidhuber**[1,2]
[1]The Swiss AI Lab IDSIA, University of Lugano (USI) & SUPSI, Lugano, Switzerland
[2]King Abdullah University of Science and Technology (KAUST), Thuwal, Saudi Arabia
{louis, juergen}@idsia.ch

## Abstract

Many concepts have been proposed for meta learning with neural networks (NNs), e.g., NNs that learn to reprogram fast weights, Hebbian plasticity, learned learning rules, and meta recurrent NNs. Our *Variable Shared Meta Learning (VSML)* unifies the above and demonstrates that simple weight-sharing and sparsity in an NN is sufficient to express powerful learning algorithms (LAs) in a reusable fashion. A simple implementation of VSML where the weights of a neural network are replaced by tiny LSTMs allows for implementing the backpropagation LA solely by running in forward-mode. It can even meta learn new LAs that differ from online backpropagation and generalize to datasets outside of the meta training distribution without explicit gradient calculation. Introspection reveals that our meta learned LAs learn through fast association in a way that is qualitatively different from gradient descent.

## 1 Introduction

The shift from standard machine learning to meta learning involves learning the learning algorithm (LA) itself, reducing the burden on the human designer to craft useful learning algorithms [43]. Recent meta learning has primarily focused on generalization from training tasks to similar test tasks, e.g., few-shot learning [11], or from training environments to similar test environments [17]. This contrasts human-engineered LAs that generalize across a wide range of datasets or environments. Without such generalization, meta learned LAs can not entirely replace human-engineered variants. Recent work demonstrated that meta learning can also successfully generate more general LAs that generalize across wide spectra of environments [20, 1, 31], e.g., from toy environments to Mujoco and Atari. Unfortunately, however, many recent approaches still rely on a large number of human-designed and unmodifiable inner-loop components such as backpropagation.

Is it possible to implement modifiable versions of backpropagation or related algorithms as part of the end-to-end differentiable activation dynamics of a neural net (NN), instead of inserting them as separate fixed routines? Here we propose the Variable Shared Meta Learning (VSML) principle for this purpose. It introduces a novel way of using sparsity and weight-sharing in NNs for meta learning. We build on the arguably simplest neural meta learner, the meta recurrent neural network (Meta RNN) [16, 10, 56], by replacing the weights of a neural network with tiny LSTMs. The resulting system can be viewed as many RNNs passing messages to each other, or as one big RNN with a sparse shared weight matrix, or as a system learning each neuron's functionality and its LA. VSML generalizes the principle behind end-to-end differentiable fast weight programmers [45, 46, 3, 41], hyper networks [14], learned learning rules [4, 13, 33], and hebbian-like synaptic plasticity [44, 46, 25, 26, 30]. Our mechanism, VSML, can implement backpropagation solely in the forward-dynamics of an RNN. Consequently, it enables meta-optimization of backprop-like algorithms. We envision a future where novel methods of credit assignment can be meta learned while still generalizing across vastly different tasks. This may lead to improvements in sample efficiency, memory efficiency, continual learning, and others. As a first step, our system meta

learns online LAs from scratch that frequently learn faster than gradient descent and generalize to datasets outside of the meta training distribution (e.g., from MNIST to Fashion MNIST). Our VSML RNN is the first neural meta learner without hard-coded backpropagation that shows such strong generalization.

## 2   Background

Deep learning-based meta learning that does not rely on fixed gradient descent in the inner loop has historically fallen into two categories, 1) Learnable weight update mechanisms that allow for changing the parameters of an NN to implement a learning rule (FWPs / LLRs), and 2) Learning algorithms implemented in black-box models such as recurrent neural networks (Meta RNNs).

**Fast weight programmers & Learned learning rules (FWPs / LLRs)**   In a standard NN, the weights are updated by a fixed LA. This framework can be extended to meta learning by defining an explicit architecture that allows for modifying these weights. This weight-update architecture augments a standard NN architecture. NNs that generate or change the weights of another or the same NN are known as fast weight programmers (FWPs) [44, 45, 46, 3, 41], hypernetworks [14], NNs with synaptic plasticity [25, 26, 30] or learned learning rules [4, 13, 33]. Often these architectures make use of local Hebbian-like update rules or outer-products, and we summarize this category as FWPs / LLRs. In FWPs / LLRs the variables $V_L$ that are subject to learning are the weights of the network, whereas the meta-variables $V_M$ that implement the LA are defined by the weight-update architecture. Note that the dimensionality of $V_L$ and $V_M$ can be defined independently of each other and often $V_M$ are reused in a coordinate-wise fashion for $V_L$ resulting in $|V_L| \gg |V_M|$, where $|\cdot|$ is the number of elements.

**Black-box learning in activations (Meta RNNs)**   It was shown that an RNN such as LSTM can learn to implement an LA [16] when the reward or error is given as an input [47]. After meta training, the LA is encoded in the weights of this RNN and determines learning during meta testing. The activations serve as the memory used for the LA solution. We refer to this as Meta RNNs [16, 10, 56] (Also sometimes referred to as memory-based meta learning.). They are conceptually simpler than FWPs / LLRs as no additional weight-update rules with many degrees of freedom need to be defined. In Meta RNNs $V_L$ are the RNN activations and $V_M$ are the parameters for the RNN. Note that an RNN with $N$ neurons will yield $|V_L| = O(N)$ and $|V_M| = O(N^2)$ [46]. This means that the LA is largely overparameterized whereas the available memory for learning is very small, making this approach prone to overfitting [20]. As a result, the RNN parameters often encode task-specific solutions instead of generic LAs. Meta learning a simple and generalizing LA would benefit from $|V_L| \gg |V_M|$. Previous approaches have tried to mend this issue by adding architectural complexity through additional memory mechanisms [53, 29, 40, 27, 42].

## 3   Variable Shared Meta Learning (VSML)

In VSML we build on the simplicity of Meta RNNs while retaining $|V_L| \gg |V_M|$ from FWPs / LLRs. We do this by reusing the same few parameters $V_M$ many times in an RNN (via variable sharing) and introducing sparsity in the connectivity. This yields several interpretations for VSML:

- (A) **VSML as a single Meta RNN with a sparse shared weight matrix (Figure 1a).** The most general description.
- (B) **VSML as message passing between RNNs (Figure 1b).** We choose a simple sharing and sparsity scheme for the weight matrix such that it corresponds to multiple RNNs with shared parameters that exchange information.
- (C) **VSML as complex neurons with learned updates (Figure 1c).** When choosing a specific connectivity between RNNs, states / activations $V_L$ of these RNNs can be interpreted as the weights of a conventional NN, consequently blurring the distinction between a weight and an activation.

**Introducing variable sharing to Meta RNNs**   We begin by formalizing Meta RNNs which often use multiplicative gates such as the LSTM [12, 15] or its variant GRU [6]. For notational simplicity,

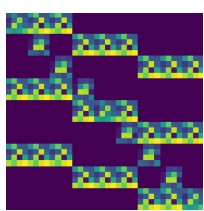

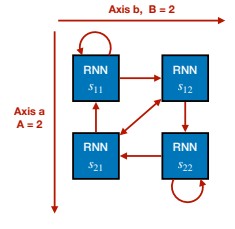

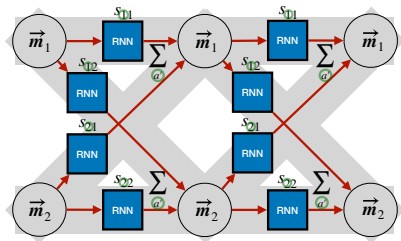

| (a) Viewed as a single RNN (structured weight matrix) | (b) One VSML RNN = many sub-RNNs | (c) Viewed as an NN with complex neurons |

Figure 1: Different perspectives on VSML: (a) A single Meta RNN [16] where entries in the weight matrix are shared or zero. (a) VSML consists of many sub-RNNs with shared parameters $V_M$ passing messages between each other. (c) VSML implements an NN with complex neurons (here 2 neurons). $V_M$ determines the nature of weights, how these are used in the neural computation, and the LA by which those are updated. Each weight $w_{ab} \in \mathbb{R}$ is represented by the multi-dimensional RNN state $s_{ab} \in \mathbb{R}^N$. Neuron activations correspond to messages $\overrightarrow{m}$ passed between sub-RNNs.

we consider a vanilla RNN. Let $s \in \mathbb{R}^N$ be the hidden state of an RNN. The update for an element $j \in \{1, \ldots, N\}$ is given by

$$s_j \leftarrow f_{\text{RNN}}(s)_j = \sigma(\sum_i s_i W_{ij}), \tag{1}$$

where $\sigma$ is a non-linear activation function, $W \in \mathbb{R}^{N \times N}$, and the bias is omitted for simplicity. We also omit inputs by assuming a subset of $s$ to be externally provided. Each application of Equation 1 reflects a single time tick in the RNN.

We now introduce variable sharing (reusing $W$) into the RNN by duplicating the computation along two axes of size $A, B$ (here $A = B$, which will later be relaxed) giving $s \in \mathbb{R}^{A \times B \times N}$. For $a \in \{1, \ldots, A\}, b \in \{1, \ldots, B\}$ we have

$$s_{abj} \leftarrow f_{\text{RNN}}(s_{ab})_j = \sigma(\sum_i s_{abi} W_{ij}). \tag{2}$$

This can be viewed as multiple RNNs arranged on a 2-dimensional grid, with shared parameters that update independent states. Here, we chose a particular arrangement (two axes) that will facilitate the interpretation Ⓒ of RNNs as weights.

**VSML as message passing between RNNs**  The computation so far describes $A \cdot B$ independent RNNs. We connect those by passing messages (interpretation Ⓑ)

$$s_{ab} \leftarrow f_{\text{RNN}}(s_{ab}, \overrightarrow{m}_a), \tag{3}$$

where the message $\overrightarrow{m}_b = \sum_{a'} f_{\overrightarrow{m}}(s_{a'b})$ with $b \in \{1, \ldots, A = B\}$, $f_{\overrightarrow{m}} : \mathbb{R}^N \to \mathbb{R}^{N'}$ is fed as an additional input to each RNN. This is related to Graph Neural Networks [51, 58]. Summing over the axis $A$ (elements $a'$) corresponds to an RNN connectivity mimicking those of weights in an NN (to facilitate interpretation Ⓒ). We emphasise that other schemes based on different kinds of message passing and graph connectivity are possible. For a simple $f_{\overrightarrow{m}}$ defined by the matrix $C \in \mathbb{R}^{N \times N}$, we may equivalently write

$$s_{abj} \leftarrow \sigma(\sum_i s_{abi} W_{ij} + \sum_{a'} f_{\overrightarrow{m}}(s_{a'a})) = \sigma(\sum_i s_{abi} W_{ij} + \sum_{a',i} s_{a'ai} C_{ij}). \tag{4}$$

This constitutes the minimal version of VSML with $V_M := (W, C)$ and is visualized in Figure 1b.

**VSML as a Meta RNN with a sparse shared weight matrix**  It is trivial to see that with $A = 1$ and $B = 1$ we obtain a single RNN and Equation 4 recovers the original Meta RNN Equation 1. In the general case, we can derive an equivalent formulation that corresponds to a single standard RNN with a single matrix $\tilde{W}$ that has entries of zero and shared entries

$$s_{abj} \leftarrow \sigma(\sum_{c,d,i} s_{cdi} \tilde{W}_{cdiabj}), \tag{5}$$

where the six axes can be flattened to obtain the two axes. For Equation 4 and Equation 5 to be equivalent, $\tilde{W}$ must satisfy (derivation in Appendix A)

$$\tilde{W}_{cdiabj} = \begin{cases} C_{ij}, & \text{if } d = a \wedge (d \neq b \vee c \neq a). \\ W_{ij}, & \text{if } d \neq a \wedge d = b \wedge c = a. \\ C_{ij} + W_{ij}, & \text{if } d = a \wedge d = b \wedge c = a. \\ 0, & \text{otherwise.} \end{cases} \tag{6}$$

This corresponds to interpretation Ⓐ with the weight matrix visualized in Figure 1a. To distinguish between the single sparse shared RNN and the connected RNNs, we now call the latter *sub-RNNs*.

**VSML as complex neurons with learned updates**    The arrangement and connectivity of the sub-RNNs as described in the previous paragraphs corresponds to that of weights in a standard NN. Thus, in interpretation Ⓒ, VSML can be viewed as defining complex neurons where each sub-RNN corresponds to a weight in a standard NN as visualized in Figure 1c. All these sub-RNNs share the same parameters but have distinct states. The current formulation corresponds to a single NN layer that is run recurrently. We will generalize this to other architectures in the next section. $A$ corresponds to the dimensionality of the inputs and $B$ to that of the outputs in that layer.

The role of weights in a standard neural network is now assigned to the states of RNNs. This allows these RNNs to define both the neural forward computation as well as the learning algorithm that determines how the network is updated (where the mechanism is shared across the network). In the case of backpropagation, this would correspond to the forward and backward passes being implemented purely in the recurrent dynamics. We will demonstrate the practical feasibility of this in Section 3.2. The emergence of

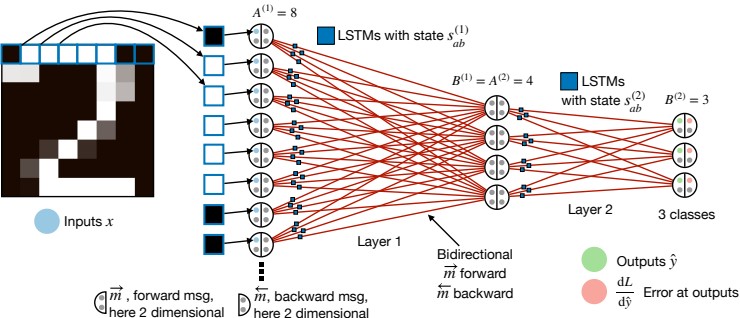

Figure 2: The neural interpretation of VSML replaces all weights of a standard NN with tiny LSTMs using shared parameters (resembling complex neurons). This allows these LSTMs to define both the neural forward computation as well as the learning algorithm that determines how the network is updated. Information flows forward and backward in the network through multi-dimensional messages $\overrightarrow{m}$ and $\overleftarrow{m}$, generalizing the dynamics of an NN trained using backpropagation.

RNN states as weights quickly leads to confusing terminology when RNNs have 'meta weights'. Instead, we simply refer to meta variables $V_M$ and learned variables $V_L$.

With this interpretation, VSML can be seen as a generalization of learned learning rules [4, 13, 33] and Hebbian-like differentiable mechanisms or fast weights more generally [44, 46, 25, 26] where RNNs replace explicit weight updates. In standard NNs, weights and activations have multiplicative interactions. For VSML RNNs to mimic such computation we require multiplicative interactions between parts of the state $s$. Fortunately, LSTMs already incorporate this through gating and can be directly used in place of RNNs.

**Stacking VSML RNNs and feeding inputs**    To get a structure similar to one of the non-recurrent deep feed-forward architectures (FNNs), we stack multiple VSML RNNs where their states are untied and their parameters are tied.[1] This is visualized with two layers in Figure 2 where the states $s^{(2)}$ of the second column of sub-RNNs are distinct from the first column $s^{(1)}$. The parameters $A^{(k)}$ and $B^{(k)}$ describing layer sizes can then be varied for each layer $k \in \{1, \dots, K\}$ constrained by $A^{(k)} = B^{(k-1)}$. The updated Equation 3 with distinct layers $k$ is given by $s_{ab}^{(k)} \leftarrow f_{\text{RNN}}(s_{ab}^{(k)}, \overrightarrow{m}_a^{(k)})$ where $\overrightarrow{m}_b^{(k+1)} := \sum_{a'} f_{\overrightarrow{m}}(s_{a'b}^{(k)})$ with $b \in \{1, \dots, B^{(k)} = A^{(k+1)}\}$.

---

[1]The resultant architecture as a whole is still recurrent. Note that even standard FNNs are recurrent if the LA (backpropagation) is taken into account.

To prevent information from flowing only forward in the network, we use an additional backward message

$$s_{ab}^{(k)} \leftarrow f_{RNN}(s_{ab}^{(k)}, \overrightarrow{m}_a^{(k)}, \overleftarrow{m}_b^{(k)}), \quad (7)$$

where $\overleftarrow{m}_a^{(k-1)} := \sum_{b'} f_{\overleftarrow{m}}(s_{ab'}^{(k)})$ with $a \in \{1, \dots, A^{(k)} = B^{(k-1)}\}$ (visualized in Figure 3). The backward transformation is given by $f_{\overleftarrow{m}} : \mathbb{R}^N \to \mathbb{R}^{N''}$.

Similarly, other neural architectures can be explicitly constructed (e.g. convolutional NNs, Section B.2). Some architectures may be learned implicitly if positional information is fed into each sub-RNN (Appendix C). We then update all states $s^{(k)}$ in sequence $1, \dots, K$ to

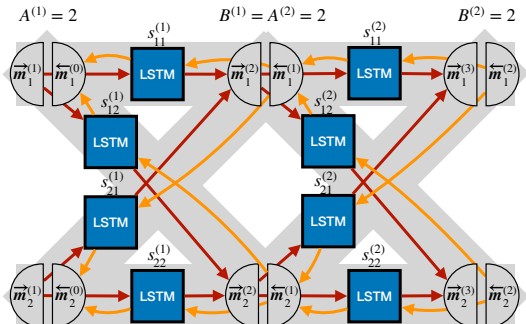

Figure 3: VSML with forward messages $\overrightarrow{m}$ and backward messages $\overleftarrow{m}$ to form a two-layer NN with shared LSTM parameters but distinct states.

mimic sequential layer execution. We may also apply multiple RNN ticks for each layer $k$.

To provide the VSML RNN with data, each time we execute the operations of the first layer, a single new datum $x \in \mathbb{R}^{A(1)}$ (e.g. one flattened image) is distributed across all sub-RNNs. In our present experiments, we match the axis $A(1)$ to the input datum dimensionality such that each dimension (e.g., pixel) is fed to different RNNs. This corresponds to initializing the forward message $\overrightarrow{m}_{a1}^{(1)} := x_a$ (padding $\overrightarrow{m}$ with zeros if necessary). Similarly, we read the output $\hat{y} \in \mathbb{R}^{B(K)}$ from $\hat{y}_a := \overrightarrow{m}_{a1}^{(K+1)}$. Finally, we feed the error $e \in \mathbb{R}^{B(K)}$ at the output such that $\overleftarrow{m}_{b1}^{(K)} := e_b$. See Figure 2 for a visualization. Alternatively, multiple input or output dimensions could be patched together and fed into fewer sub-RNNs.

### 3.1 Meta learning general-purpose learning algorithms from scratch

Having formalized VSML, we can now use end-to-end meta learning to create LAs from scratch in Algorithm 1. We simply optimize the LSTM parameters $V_M$ to minimize the sum of prediction losses over many time steps starting with random states $V_L := \{s_{ab}^{(k)}\}$. We focus on meta learning online LAs where one example is fed at a time as done in Meta RNNs [16, 56, 10]. Meta training may be performed using end-to-end gradient descent or gradient-free optimization such as evolutionary strategies [57, 38]. The latter is significantly more efficient on VSML compared to standard NNs due to the small parameter space $V_M$. Crucially, during meta testing, no explicit gradient descent is used.

---

**Algorithm 1** VSML: Meta Training

---

**Require:** Dataset(s) $D = \{(x_i, y_i)\}$
  $V_M \leftarrow$ initialize LSTM parameters
  **while** meta loss has not converged **do**          ▷ Outer loop in parallel over datasets $D$
      $V_L = \{s_{ab}^{(k)}\} \leftarrow$ initialize LSTM states    $\forall a, b, k$
      **for** $(x, y) \in \{(x_1, y_1), \dots, (x_T, y_T)\} \subset D$ **do**          ▷ Inner loop over $T$ examples
        $\overrightarrow{m}_{a1}^{(1)} := x_a$    $\forall a$          ▷ Initialize from input image x
        **for** $k \in \{1, \dots, K\}$ **do**          ▷ Iterating over $K$ layers
            $s_{ab}^{(k)} \leftarrow f_{RNN}(s_{ab}^{(k)}, \overrightarrow{m}_a^{(k)}, \overleftarrow{m}_b^{(k)})$    $\forall a, b$          ▷ Equation 7
            $\overrightarrow{m}_b^{(k+1)} := \sum_{a'} f_{\overrightarrow{m}}(s_{a'b}^{(k)})$    $\forall b$          ▷ Create forward message
            $\overleftarrow{m}_a^{(k-1)} := \sum_{b'} f_{\overleftarrow{m}}(s_{ab'}^{(k)})$    $\forall a$          ▷ Create backward message
        $\hat{y}_a := \overrightarrow{m}_{a1}^{(K+1)}$    $\forall a$          ▷ Read output
        $e := \nabla_{\hat{y}} L(\hat{y}, y)$          ▷ Compute error at outputs using loss $L$
        $\overleftarrow{m}_{b1}^{(K)} := e_b$    $\forall b$          ▷ Input errors
      $V_M \leftarrow V_M - \alpha \nabla_{V_M} \sum_{t=1}^{T} L(\hat{y}(t), y(t))$, obtaining $\nabla_{V_M}$ either by
       • back-propagation through the inner loop
       • evolution strategies, using a search distribution $p(V_M)$

---

## 3.2 Learning to implement backpropagation in RNNs

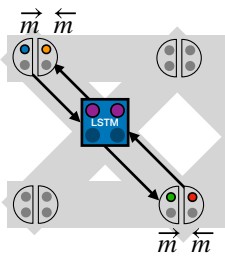

Figure 4: To implement back-propagation we optimize the VSML RNN to use and update weights $w$ and biases $b$ as part of the state $s_{ab}$ in each sub-RNN. Inputs are pre-synaptic $x$ and error $e$. Outputs are post-synaptic $\hat{y}$ and error $\hat{e}'$.

An alternative to end-to-end meta learning is to first investigate whether the VSML RNN can implement backpropagation. Due to the algorithm's ubiquitous use, it seems desirable to be able to meta learn backpropagation-like algorithms. Here we investigate how VSML RNNs can learn to implement backpropagation purely in their recurrent dynamics. We do this by optimizing $V_M$ to (1) store a weight $w$ and bias $b$ as a subset of each state $s_{ab}$, (2) compute $y = \tanh(x)w + b$ to implement neural forward computation, and (3) update $w$ and $b$ according to the backpropagation algorithm [23]. We call this process *learning algorithm cloning* and it is visualized in Figure 4.

We designate an element of each message $\overrightarrow{m}_a^{(k)}$, $\overleftarrow{m}_b^{(k)}$, $f_{\overrightarrow{m}}(s_{ab}^{(k)})$, $f_{\overleftarrow{m}}(s_{ab}^{(k)})$ as the input $x$, error $e$, and output $\hat{y}$ and error $\hat{e}'$. Similarly, we set $w := s_{ab1}$ and $b := s_{ab2}$. We then optimize $V_M$ via gradient descent to regress $\hat{y}$, $\Delta w$, $\Delta b$, and $\hat{e}'$ toward their respective targets. We can either generate the training dataset $D := \{(x, w, b, y, e, e')_i\}$ randomly or run a 'shadow' NN on some supervised problem and fit the VSML RNN to its activations and parameter updates. Multiple iterations in the VSML RNN would then correspond to evaluating the network and updating it via backpropagation. The activations from the forward pass necessary for credit assignment could be memorized as part of the state $s$ or be explicitly stored and fed back. For simplicity, we chose the latter to clone backpropagation. We continuously run the VSML RNN forward, alternately running the layers in order $1, \ldots, K$ and in reverse order $K, \ldots, 1$.[2]

## 4 Experiments

First, we demonstrate the capabilities of the VSML RNN by showing that it can implement neural forward computation and backpropagation in its recurrent dynamics on the *MNIST* [21] and *Fashion MNIST* [59] dataset. Then, we show how we can meta learn an LA from scratch on one set of datasets and then successfully apply it to another (out of distribution). Such generalization is enabled by extensive variable sharing where we have very few meta variables $|V_M| \approx 2,400$ and many learned variables $|V_L| \approx 257,200$. We also investigate the robustness of the discovered LA. Finally, we introspect the meta learned LA and compare it to gradient descent.

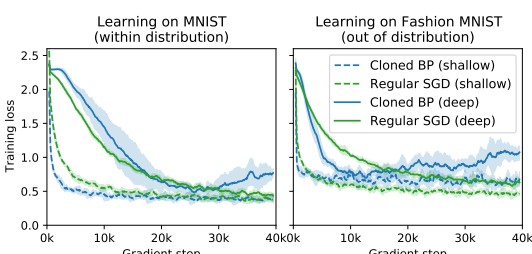

Figure 5: The VSML RNN is optimized to implement backpropagation in its recurrent dynamics on MNIST, then tested both on MNIST and Fashion MNIST where it performs learning purely by unrolling the LSTM. We test on shallow and deep architectures (single hidden layer of 32 units). Standard deviations are over 6 seeds.

Our implementation uses LSTMs and the message interpretation from Equation 7. Hyperparameters, training details, and additional experiments can be found in the appendix.

### 4.1 VSML RNNs can implement backpropagation

As described in Section 3.2, we optimize the VSML RNN to implement backpropagation. We structure the sub-RNNs to mimic a feed-forward NN with either one hidden layer or no hidden layers. To obtain training targets, we instantiate a standard NN, the shadow network, and feed it MNIST data. After cloning, we then run the LA encoded in the VSML RNN on the MNIST and Fashion MNIST dataset and observe that it performs learning purely in its recurrent dynamics, making explicit gradient calculations unnecessary. Figure 5 shows the learning curve on these two datasets. Notably,

---

[2]Executing layers in reverse order is not strictly necessary as information always also flows backwards through $\overleftarrow{m}$ but makes LA cloning easier.

learning works both on MNIST (within distribution) and on Fashion MNIST (out of distribution). We observe that the loss is decently minimized, albeit regular gradient descent still outperforms our cloned backpropagation. This may be due to non-zero errors during learning algorithm cloning, in particular when these errors accumulate in the deeper architecture. It is also possible that the VSML states ('weights') deviate too far from ranges seen during cloning, in particular in the deep case when the loss starts increasing. We obtain 87% (deep) and 90% (shallow) test accuracy on MNIST and 76% (deep) and 80% (shallow) on Fashion MNIST (focusing on successful cloning over performance).

## 4.2 Meta learning supervised learning from scratch

In the previous experiments, we have established that VSML is expressive enough to meta-optimize backpropagation-like algorithms. Instead of cloning an LA, we now meta learn from scratch as described in Section 3.1. Here, we use a single layer ($K = 1$) from input to output dimension and run it for two ticks per image with $N = 16$ and $N' = N'' = 8$. First, the VSML RNN is meta trained end-to-end using evolutionary strategies (ES) [38] on MNIST to minimize the sum of cross-entropies over 500 data points starting from random state initializations. As each image is unique and $V_M$ can not memorize the data, we are implicitly optimizing the VSML RNN to generalize to future inputs given all inputs it has seen so far. We do not pre-train $V_M$ with a human-engineered LA.

During meta testing on MNIST (Figure 6) we plot the cumulative accuracy on all previous inputs on the y axis ($\frac{1}{T}\sum_{t=1}^{T} c_t$ after example $T$

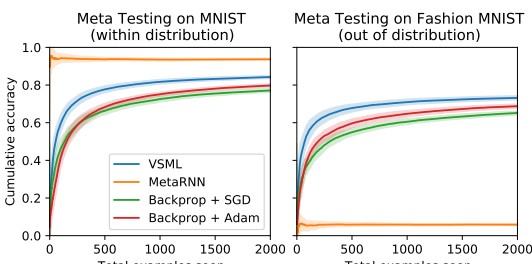

Figure 6: The VSML RNN can be directly meta trained on MNIST to minimize the sum of errors when classifying online starting from a random state initialization. This allows for faster learning during meta testing compared to online gradient descent with Adam on the same dataset and even generalizes to a different dataset (Fashion MNIST). In comparison, a standard Meta RNN [16] strongly overfits in the same setting. Standard deviations are over 128 seeds.

with binary $c_t$ indicating prediction correctness). For each example, the prediction when this example was fed to the RNN is used, thus measuring sample efficient learning. This evaluation protocol is similar to the one used in Meta RNNs [56, 10]. We observe that learning is considerably faster compared to the baseline of online gradient descent (no mini batching, the learning rate appropriately tuned). One possibility is that VSML simply overfits to the training distribution. We reject this possibility by meta testing the same unmodified RNN on a different dataset, here Fashion MNIST. Learning still works well, meaning we have meta learned a fairly general LA (although performance at convergence still lacks behind a little). This generalization is achieved without using any hardcoded gradients during meta testing purely by running the RNN forward. In comparison to VSML, a Meta RNN heavily overfits.

## 4.3 Robustness to varying inputs and outputs

A defining property of VSML is that the same parameters $V_M$ can be used to learn on varying input and output sizes. Further, the architecture and thus the meta learned LA is invariant to the order of inputs and outputs. In this experiment, we investigate how robust we are to such changes. We meta train across MNIST with 3, 4, 6, and 7 classes. Likewise, we train across rescaled versions with 14x14, 28x28, and 32x32 pixels. We also randomly project all inputs using a linear transformation, with the transformation fixed for all inner learning steps. In Figure 7 we meta test on 6 configurations that were not seen

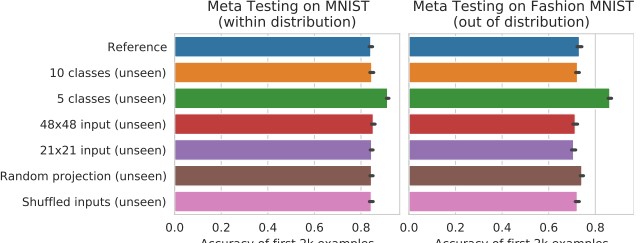

Figure 7: The meta learned learning algorithm is robust to permutations and size changes in the inputs and outputs. All six configurations have not been seen during training and perform comparable to the unchanged reference. Standard deviations are over 32 seeds.

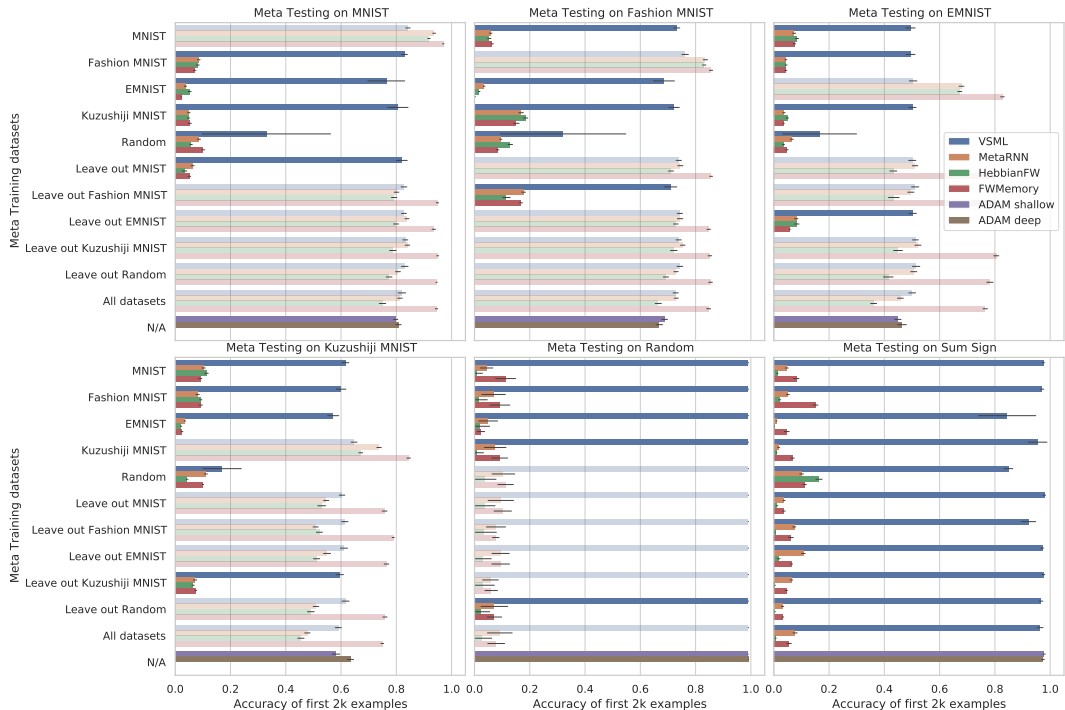

Figure 8: Online learning on various datasets. Cumulative accuracy in % after having seen 2k training examples evaluated after each prediction starting with random states (VSML, Meta RNN, HebbianFW, FWMemory) or random parameters (SGD). Standard deviations are over 32 meta test training runs. Meta testing is done on the official test set of each dataset. Meta training is on subsets of datasets excluding the Sum Sign dataset. Unseen tasks, most relevant from a general-purpose LA perspective, are opaque.

during meta training. Performance on all of these configurations is comparable to the unchanged reference from the previous section. In particular, the invariance to random projections suggests that we have meta learned a learning algorithm beyond transferring learned representations [compare 11, 54, 55].

## 4.4 Varying datasets

To better understand how different meta training distributions and meta test datasets affect VSML RNNs and our baselines, we present several different combinations in Figure 8. The opaque bars represent tasks that were not seen during meta training and are thus most relevant for this analysis. This includes four additional datasets: (1) *Kuzushiji MNIST* [7] with 10 classes, (2) *EMNIST* [9] with 62 classes, (3) A randomly generated classification dataset (*Random*) with 20 data points that changes with each step in the outer loop, and (4) *Sum Sign* which generates random inputs and requires classifying the sign of the sum of all inputs. Meta training is done over 500 randomly drawn samples per outer iteration. Each algorithm is meta trained for 10k outer iterations. Inputs are randomly projected as in Section 4.3 (for VSML; the baselines did not benefit from these augmentations). We again report the cumulative accuracy on all data seen since the beginning of meta test training. We compare to SGD with a single layer, matching the architecture of VSML, and a hidden layer, matching the number of weights to the size of $V_L$ in VSML. We also have included a Hebbian fast weight baseline [25] and an external (fast weight) memory approach [42].

We observe that VSML generalizes much better than Meta RNNs, Hebbian fast weights, and the external memory. These baselines overfit to the training environments. Notably, VSML even generalizes to the unseen tasks *Random* and *Sum Sign* which have no shared structure with the other datasets. In many cases VSML's performance is similar to SGD but a little more sample efficient in the beginning of training (learning curves in Appendix B). This suggests that our meta learned LAs are good at quickly associating new inputs with their labels. We further investigate this in the next Section 5.

## 5 Analysis

Given that VSML seems to learn faster than online gradient descent in many cases we would like to qualitatively investigate how learning differs. We first meta train on the full MNIST dataset as before. During meta testing, we plot the output probabilities for each digit against the number of samples seen in Figure 9. We highlight the ground truth input class □ as well as the predicted class ◯. In this case, our meta test dataset consists of MNIST digits with two examples of each type. The same digit is always repeated twice. This allows us to observe and visualize the effect with only a few examples. We have done the same introspection with the full dataset in Appendix B.

We observe that in VSML almost all failed predictions are followed by the correct prediction with high certainty. In contrast, SGD

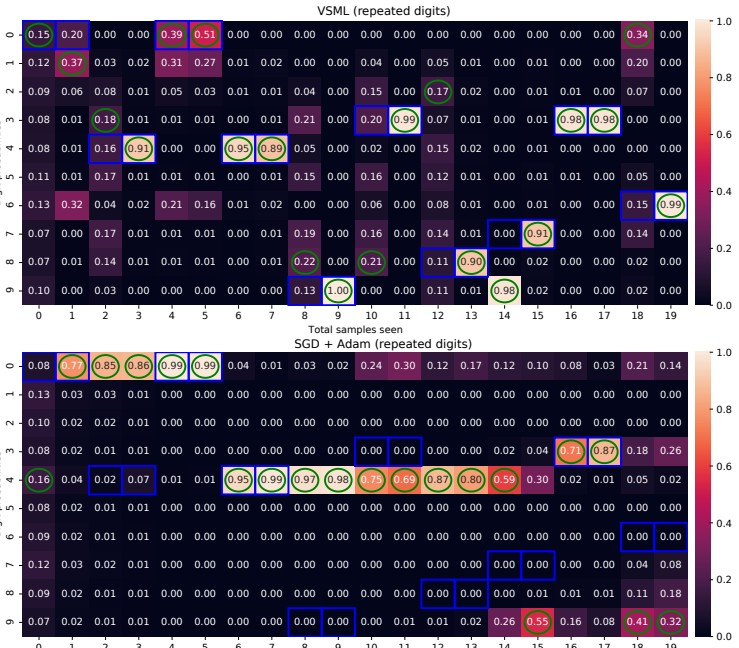

Figure 9: Introspection of how output probabilities change after observing an input and the prediction error when meta testing on MNIST with two examples for each type. We highlight the ground truth class □ as well as the predicted class ◯. The top plot shows VSML quickly associating the input images with the right label, almost always making the right prediction the second time with high confidence. The bottom plot shows the same dataset processed by SGD with Adam which fails to learn quickly.

makes many incorrect predictions and fails to adapt correctly in only 20 steps. It seems that SGD learns qualitatively different from VSML. The VSML RNN meta learns to quickly associate new inputs with their class whereas SGD fails to do so. We tried several different SGD learning rates and considered multiple steps on the same input. In both cases, SGD does not behave similar to VSML, either learning much slower or forgetting previous examples. As evident from high accuracies in Figure 8, VSML does not only memorize inputs using this strategy of fast association, but the associations generalize to future unseen inputs.

## 6 Related Work

**Memory based meta learning (Meta RNNs)** Memory-based meta learning in RNNs [16, 10, 56] is a simple neural meta learner (see Section 2). Unfortunately, the LA encoded in the RNN parameters is largely overparameterized which leads to overfitting. VSML demonstrates that weight sharing can address this issue resulting in more general-purpose LAs.

**Learned Learning Rules / Fast Weights** NNs that generate or change the weights of another or the same NN are known as fast weight programmers [44], hypernetworks [14], NNs with synaptic plasticity [25] or learned learning rules [4] (see Section 2). In VSML we do not require explicit architectures for weight updates as weights are emergent from RNN state updates. In addition to the learning rule, we implicitly learn how the neural forward computation is defined. Concurrent to this work, fast weights have also been used to meta learn more general LAs [39].

**Learned gradient-based optimizers** Meta learning has been used to find optimizers that update the parameters of a model by taking the loss and gradient with respect to these parameters as an input [34, 2, 22, 24]. In this work, we are interested in meta learning that does not rely on fixed gradient calculation in the inner loop.

**Discrete program search** An interesting alternative to distributed variable updates in VSML is meta learning via discrete program search [48, 35]. In this paradigm, a separate programming language needs to be defined that gives rise to neural computation when its instructions are combined. This led to the automated rediscovery of backpropagation [35]. In VSML we demonstrate that a symbolic programming language is not required and general-purpose LAs can be discovered and encoded in variable-shared RNNs. Search over neural network parameters is usually easier compared to symbolic program search due to smoothness in the loss landscape.

**Multi-agent systems** In the reinforcement learning setting multiple agents can be modeled with shared parameters [50, 32, 18], also in the context of meta learning [36]. This is related to the variable sharing in VSML depending on how the agent-environment boundary is drawn. Unlike these works, we demonstrate the advantage of variable sharing in meta learning more general-purpose LAs and present a weight update interpretation.

## 7    Discussion and Limitations

The research community has perfected the art of leveraging backpropagation for learning for a long time. At the same time, there are open questions such as how to minimize memory requirements, effectively learn online and continually, learn sample efficiently, learn without separate backward phases, and others. VSML suggests that instead of building on top of backpropagation as a fixed routine, meta learning offers an alternative to discover general-purpose LAs. Nevertheless, this paper is only a proof of concept—until now we have only investigated small-scale problems and performance does not yet quite match the mini-batched setting with large quantities of data. In particular, we observed premature convergence of the solution at meta test time which calls for further investigations. Scaling our system to harder problems and larger meta task distributions will be important future work.

The computational cost of the current VSML variant is also larger than the one of standard back-propagation. If we run a sub-RNN for each weight in a standard NN with $W$ weights, the cost is in $O(WN^2)$, where $N$ is the state size of a sub-RNN. If $N$ is small enough, and our experiments suggest small $N$ may be feasible, this may be an acceptable cost. However, VSML is not bound to the interpretation of a sub-RNN as one weight. Future work may relax this particular choice.

Meta optimization is also prone to local minima. In particular, when the number of ticks between input and feedback increases (e.g. deeper architectures), credit assignment becomes harder. Early experiments suggest that diverse meta task distributions can help mitigate these issues. Additionally, learning horizons are limited when using backprop-based meta optimization. Using ES allowed for training across longer horizons and more stable optimization.

VSML can also be viewed as regularizing the NN weights that encode the LA through a representational bottleneck. It is conceivable that LA generalization as obtained by VSML can also be achieved through other regularization techniques. Unlike most regularizers, VSML also introduces substantial reuse of the same learning principle and permutation invariance through variable sharing.

## 8    Conclusion

We introduced Variable Shared Meta Learning (VSML), a simple principle of weight sharing and sparsity for meta learning powerful learning algorithms (LAs). Our implementation replaces the weights of a neural network with tiny LSTMs that share parameters. We discuss connections to meta recurrent neural networks, fast weight generators (hyper networks), and learned learning rules.

Using *learning algorithm cloning*, VSML RNNs can learn to implement the backpropagation algorithm and its parameter updates encoded implicitly in the recurrent dynamics. On MNIST it learns to predict well without any human-designed explicit computational graph for gradient calculation.

VSML can meta learn from scratch supervised LAs that do not explicitly rely on gradient computation and that *generalize to unseen datasets*. Introspection reveals that VSML LAs learn by fast association in a way that is qualitatively different from stochastic gradient descent. This leads to gains in sample efficiency. Future work will focus on reinforcement learning settings, improvements of meta learning, larger task distributions, and learning over longer horizons.

## Acknowledgements

We thank Sjoerd van Steenkiste, Imanol Schlag, Kazuki Irie, and the anonymous reviewers for their comments and feedback. This work was supported by the ERC Advanced Grant (no: 742870) and computational resources by the Swiss National Supercomputing Centre (CSCS, projects s978 and s1041). We also thank NVIDIA Corporation for donating several DGX machines as part of the Pioneers of AI Research Award, IBM for donating a Minsky machine, and weights & biases [5] for their great experiment tracking software and support.

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
