# A   Derivations

**Theorem 1.** *The weight matrices $W$ and $C$ used to compute VSML RNNs from Equation 4 can be expressed as a standard RNN with weight matrix $\tilde{W}$ (Equation 5) such that*

$$s_{abj} \leftarrow \sigma(\sum_i s_{abi} W_{ij} + \sum_{c,i} s_{cai} C_{ij}) \tag{8}$$

$$= \sigma(\sum_{c,d,i} s_{cdi} \tilde{W}_{cdiabj}). \tag{9}$$

*The weight matrix $\tilde{W}$ has entries of zero and shared entries given by Equation 6.*

$$\tilde{W}_{cdiabj} = \begin{cases} C_{ij}, & \text{if } d = a \wedge (d \neq b \vee c \neq a). \\ W_{ij}, & \text{if } d \neq a \wedge d = b \wedge c = a. \\ C_{ij} + W_{ij}, & \text{if } d = a \wedge d = b \wedge c = a. \\ 0, & \text{otherwise.} \end{cases} \tag{6 revisited}$$

*Proof.* We rearrange $\tilde{W}$ into two separate weight matrices

$$\sum_{c,d,i} s_{cdi} \tilde{W}_{cdiabj} \tag{10}$$

$$= \sum_{c,d,i} s_{cdi} A_{cdiabj} + \sum_{c,d,i} s_{cdi} (\tilde{W} - A)_{cdiabj}. \tag{11}$$

Then assuming $A_{cdiabj} = (d \equiv b)(c \equiv a) W_{ij}$, where $x \equiv y$ equals 1 iff $x$ and $y$ are equal and 0 otherwise, it holds that

$$\sum_{c,d,i} s_{cdi} A_{cdiabj} = \sum_i s_{abi} W_{ij}. \tag{12}$$

Further, assuming $(\tilde{W} - A)_{cdiabj} = (d \equiv a) C_{ij}$ we obtain

$$\sum_{c,d,i} s_{cdi} (\tilde{W} - A)_{cdiabj} = \sum_{c,i} s_{cai} C_{ij}. \tag{13}$$

Finally, solving both conditions for $\tilde{W}$ gives

$$\tilde{W}_{cdiabj} = (d \equiv a) C_{ij} + (d \equiv b)(c \equiv a) W_{ij}, \tag{14}$$

which we rewrite in tabular notation:

$$\tilde{W}_{cdiabj} = \begin{cases} C_{ij}, & \text{if } d = a \wedge (d \neq b \vee c \neq a). \\ W_{ij}, & \text{if } d \neq a \wedge d = b \wedge c = a. \\ C_{ij} + W_{ij}, & \text{if } d = a \wedge d = b \wedge c = a. \\ 0, & \text{otherwise.} \end{cases} \tag{15}$$

Thus, Equation 8 holds and any weight matrices $W$ and $C$ can be expressed by a single weight matrix $\tilde{W}$. $\qquad\square$

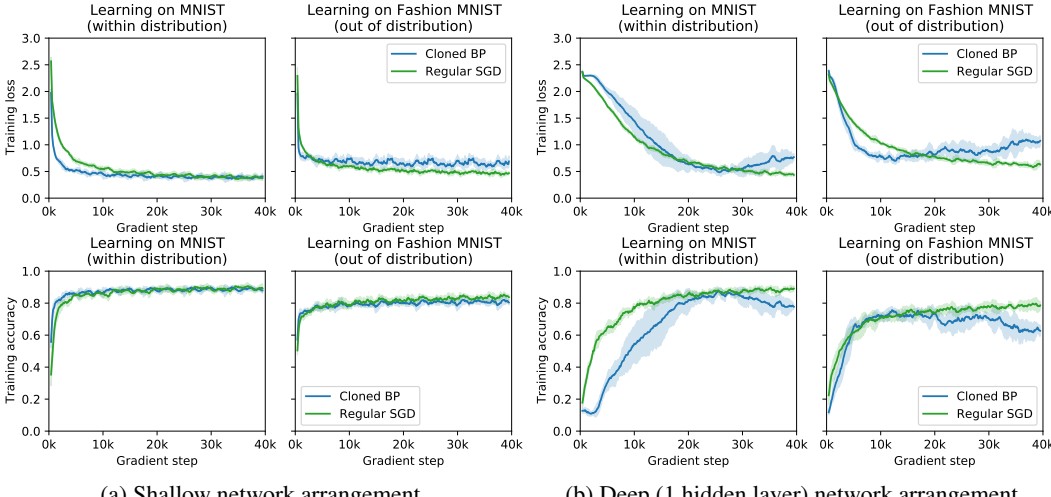

(a) Shallow network arrangement.  (b) Deep (1 hidden layer) network arrangement.

Figure 11: Additional experiments with VSML RNNs implementing backpropagation. Standard deviations are over 6 seeds.

# B   Additional Experiments

## B.1   Learning algorithm cloning

**VSML RNNs can implement neural forward computation**    In this experiment, we optimize the VSML RNN to compute $y = tanh(x)w$. Figure 10 (left) shows how for different inputs $x$ and weights $w$ the LSTM produces the correct target value, including the multiplicative interaction. The heat-map (right) shows that low prediction errors are produced but the target dynamics are not perfectly matched. We repeat these LSTMs in line with Equation 7 to obtain an 'emergent' neural network.

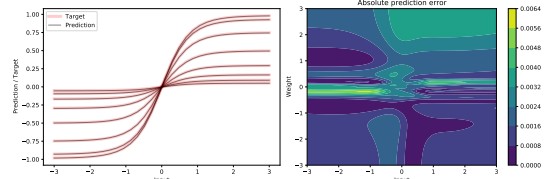

Figure 10: We are optimizing VSML RNNs to implement neural forward computation such that for different inputs and weights a tanh-activated multiplicative interaction is produced (left), with different lines for different $w$. These neural dynamics are not exactly matched everywhere (right), but the error is relatively small.

**Learning Algorithm Cloning Curriculum** In principle, backpropagation can be simply cloned on random data such that forward computation implements multiplicative activation-weight interaction and backward computation passes an error signal back given previous forward activations. If the previous forward activations are fed as an input one could stack VSML RNNs that implement these two operations to mimic arbitrarily deep NNs. By purely training on random data and unrolling for one step, we can successfully learn on MNIST and Fashion MNIST in the shallow setting. For deeper models, in practice, cloning errors accumulate and input and state distributions shift. To achieve learning in deeper networks we have used a curriculum on random and MNIST data. We first match the forward activations, backward errors, and weight updates for a shallow network. Next, we use a deep network and provide intermediate errors by a ground truth network. Finally, we remove intermediate errors and use the RNN's intermediate predictions that are now close to the ground truth. The final VSML RNN can be used to train both shallow (Figure 11a) and deep configurations (Figure 11b).

## B.2   Meta learning from scratch

**Meta testing learning curves & sample efficiency**    In Figure 8 we only showed accuracies after 2k steps. Figure 12 provides the entire meta test training trajectories for a subset of all configurations. Furthermore, in Figure 13 we show the cumulative accuracy on the first 100 examples. From both

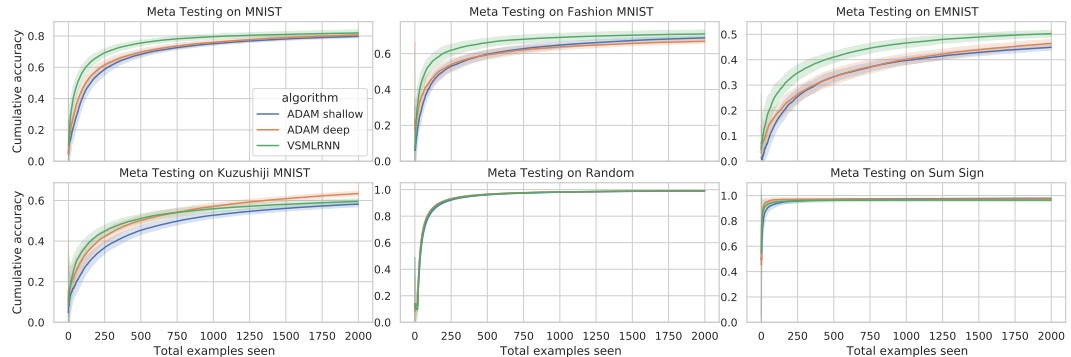

Figure 12: Meta testing learning curves. All 6 meta test tasks are unseen. VSML RNN has been meta trained on MNIST, Fashion MNIST, EMNIST, KMNIST, and Random, excluding the respective dataset that is being meta tested on. Standard deviations are over 32 seeds.

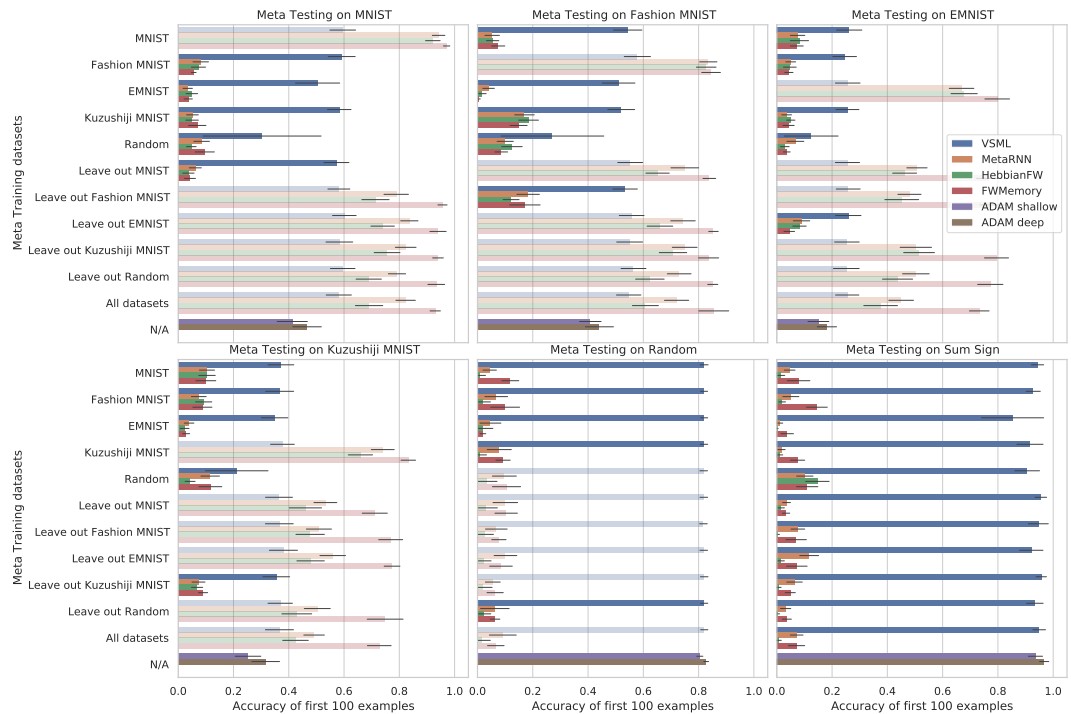

Figure 13: Online learning on various datasets. Cumulative accuracy in % after having seen **100 training examples** evaluated after each prediction starting with random states (VSML, Meta RNN, HebbianFW, FWMemory) or random parameters (SGD). Standard deviations are over 32 meta test training runs. Meta testing is done on the official test set of each dataset. Meta training is on subsets of datasets excluding the Sum Sign dataset. Unseen tasks, most relevant from a general-purpose LA perspective, are opaque.

figures, it is evident that learning at the beginning is accelerated compared to SGD with Adam. Also compare with our introspection from Section 5.

**Ablation: Projection augmentations** In the main text (Figure 8) we have randomly projected inputs during VSML meta training. When not randomly projecting inputs (Figure 14), generalization of VSML is slightly reduced. In Figure 15 we have enabled these augmentations for all methods, including the baselines. While VSML benefits from the augmentations, the Meta RNN, Hebbian fast weights, and external memory baselines do not increase their generalization significantly with

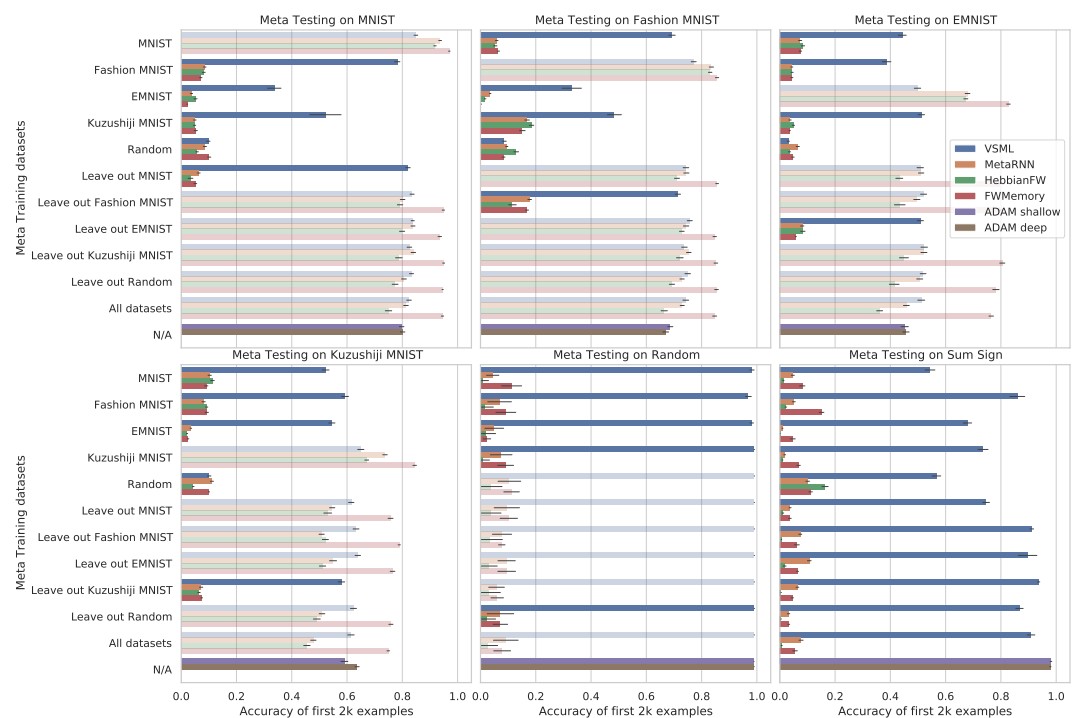

Figure 14: Same as figure Figure 8 and Figure 13 but with accuracies after having seen 2k training examples and **no random projections for all methods** during meta training.

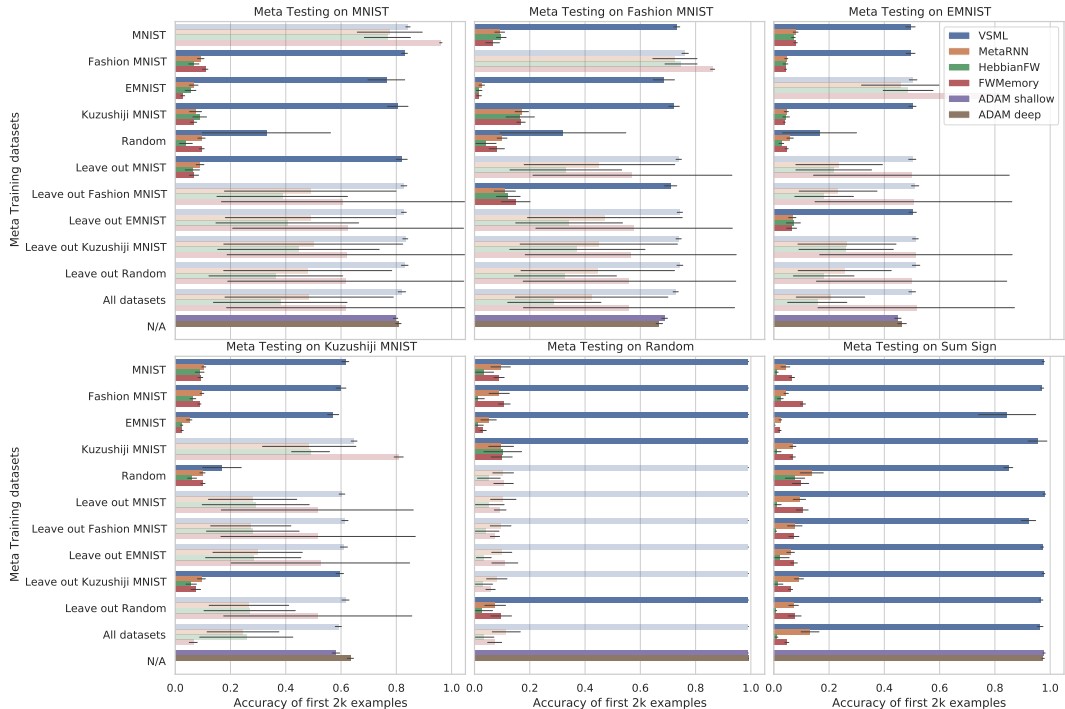

Figure 15: Same as figure Figure 8 and Figure 13 but with accuracies after having seen 2k training examples and **random projections for all methods including baselines** during meta training.

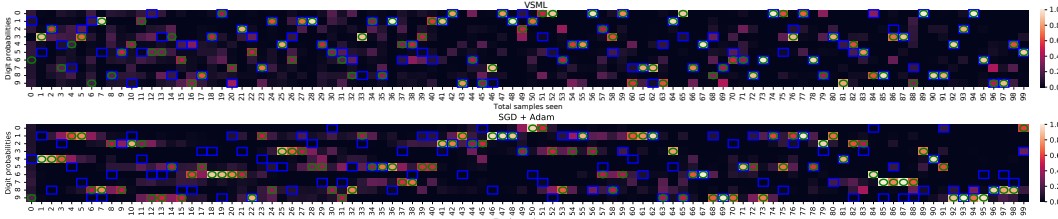

Figure 17: Introspection of how output probabilities change after observing an input and its error at the output units when meta testing **on the full MNIST dataset**. We highlight the input class □ as well as the predicted class ◯ for 100 examples in sequence. The top plot shows the VSML RNN quickly associating the input images with the right label, generalizing to future inputs. The bottom plot shows the same dataset processed by SGD with Adam which learns significantly slower by following the gradient.

those enabled. In Figure 16 we show meta test training curves for both the augmented as well as non-augmented case.

**Introspect longer meta test training run** Similar to Figure 9 we introspect how VSML RNNs learned to learn after meta training on the MNIST dataset. In this case, we meta test for 100 steps by sampling from the full MNIST dataset in Figure 17 without repeating digits. Compared to the previous setup, learning is slower as there is a larger variety of possible inputs. Nevertheless, we observe that VSML RNNs still associate inputs with their label rather quickly compared to SGD.

**Omniglot** In this paper, we have focused on the objective of meta learning a general-purpose

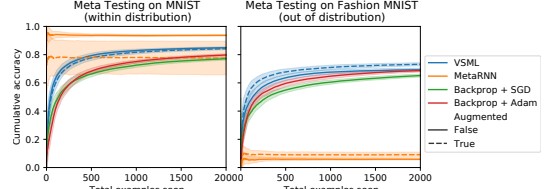

Figure 16: On the MNIST meta training example from Figure 6 we plot the effect of adding the random projection augmentation to VSML and the Meta RNN. The Fashion MNIST performance (out of distribution) is slightly improved for VSML while the effect on the Meta RNN is limited.

learning algorithm. Different from most contemporary meta learning approaches we tested the discovered learning algorithm on significantly different datasets to assess its generalization capabilities. These generalization capabilities may affect the performance on standard few-shot benchmarks such as Omniglot. In this section, we assess how VSML performs on those datasets where the tasks at meta test time are similar to those during meta training.

On Omniglot, our experimental setting corresponds to the common 5-way, 1-shot setting [25]: In each episode, we select 5 random classes, sample 1 instance each, and show it to the network with the label and prediction error. Then, we sample a new random test instance from one of the 5 classes and meta train to minimize the cross-entropy on that example. At meta test time we use unseen alphabets (classes) from the test set and report the accuracy of the test instance across 100 episodes.

The results (Figure 18) nicely demonstrate how common baselines such as the Meta RNN [16, 10, 56] or a Meta RNN with external memory [42] work well in an Omniglot setting, yet fail when the gap increases between meta train

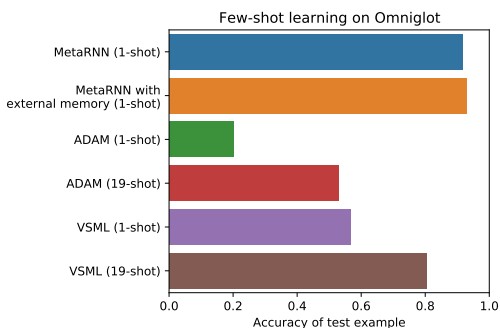

Figure 18: VSML on the Omniglot dataset.

and meta test, thus requiring stronger generalization (Figure 6, Figure 8). In contrast, VSML generalizes well to unseen datasets, e.g. Fashion MNIST, although it does learn more slowly on Omniglot. Finally, these new results demonstrate how VSML learns significantly faster on Omniglot compared to SGD with Adam, thus highlighting the benefits of the meta learning approach adopted in this work.

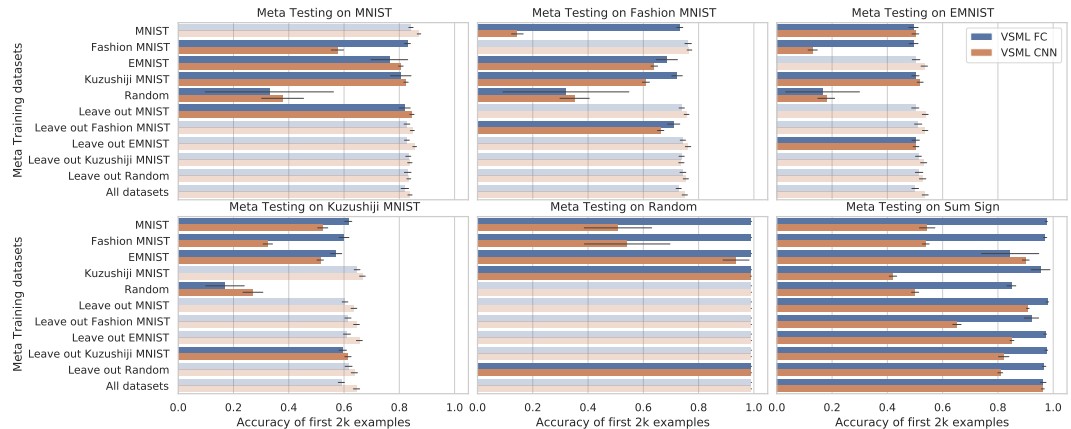

Figure 20: Convolutions are competitive to the standard fully connected setup.

**Short horizon bias**  In this paper, we have observed that VSML can be significantly more sample efficient compared to backpropagation with gradient descent, in particular for the first few examples. The longer we unroll the VSML RNNs, the smaller this gap becomes. In Figure 19 we run VSML for $12,000$ examples ($24,000$ RNN ticks). From this plot, it is evident that at some point gradient descent overtakes VSML in terms of learning progress. We call this phenomenon the *short horizon bias*, where meta test training is fast in the beginning but flattens out at some horizon.

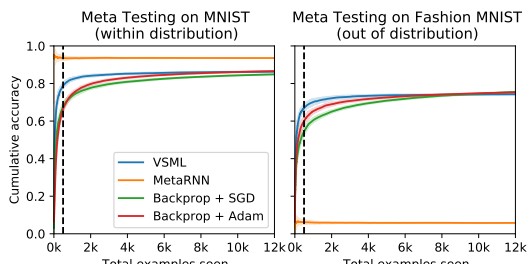

Figure 19: Short horizon bias.

In the current version of VSML we only meta optimize the RNN for 500 examples (marked by the vertical dashed line) starting with a random initialization, not explicitly optimizing learning beyond that point, resulting in this bias. In future work, we will investigate methods to circumvent this bias, for example by resuming from previous states (learning progress) similar to a persistent population in previous meta learning work [20].

**Convolutional Neural Networks**  VSML's sub-RNNs can not only be arranged to fully connected layers, but also convolutions. For this experiment, we have implemented a convolutional neural network (CNN) version of VSML. This is done by replacing each weight in the kernel with a multi-dimensional RNN state and replacing the kernel multiplications with VSML sub-RNNs. We use a convolutional layer with kernel size 3, stride 2, and 8 channels, followed by a dense layer. On our existing datasets, it performs similar to the fully connected architecture, as can be seen in Figure 20.

We also applied our CNN variant to CIFAR10. Note that in this paper we are interested in the online learning setting (similar to the one of Meta RNNs). This is a challenging problem on which gradient descent with back-propagation also struggles. Many consecutive examples ($> 10^5$ steps) are required for learning. Online performance is generally lower than in the batched setting which we do not explore here. When meta training on CIFAR10 (Figure 21) we observe that meta test-time learning on CIFAR is initially faster compared to SGD while still generalizing to Fashion MNIST. On the other hand, with a sufficiently large meta training dis-

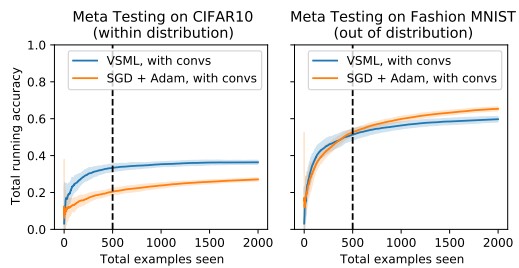

Figure 21: Meta Training on CIFAR10 with a CNN version of VSML.

tribution, we would hope to see a similar generalization to CIFAR10 when CIFAR10 is unseen. As visible in both plots, learning speed decreases at some point. This is probably due to the current

short-horizon bias as discussed in the previous paragraph. Future improvements are necessary to further scale VSML to harder learning problems.

## C  Other Training Details

**LSTM implementation**   We implement the VSML RNN using $A \cdot B$ LSTMs with forward and backward messages as described in Equation 7. Each LSTM $ab$ at layer $k$ is updated by

$$z_{ab}^{(k)}, h_{ab}^{(k)} \leftarrow f_{\text{LSTM}}(z_{ab}^{(k)}, h_{ab}^{(k)}, \overrightarrow{m}_a^{(k)}, \overleftarrow{m}_b^{(k)}). \tag{16}$$

The functions $f_{\overrightarrow{m}}$ and $f_{\overleftarrow{m}}$ are a linear projection to outputs of size $N' = 8$ and $N'' = 8$ respectively. The state size is given by $N = 64$ for LA cloning and $N = 16$ for meta learning from scratch. $A^{(1)}$ and $B^{(K)}$ are fixed according to the dataset input/output size and others are chosen freely as described in the respective experiment. We found that averaging messages instead of summing them, $\overrightarrow{m}_b^{(k)} := \frac{1}{A^{(k-1)}} \sum_{a'} f_{\overrightarrow{m}}(s_{a'b}^{(k-1)})$ and $\overleftarrow{m}_a^{(k)} := \frac{1}{B^{(k+1)}} \sum_{b'} f_{\overleftarrow{m}}(s_{ab'}^{(k+1)})$, improves meta training stability.

**Source code** is available at `http://louiskirsch.com/code/vsml`.

### C.1  Learning algorithm cloning

**General training remarks**   During the forward evaluation of layers $1, \ldots, K$ we freeze the LSTM state. During the backward pass, we only retain two state dimensions that correspond to the weight and the bias. We also zero all other LSTM input dimensions in $\overrightarrow{m}$ and $\overleftarrow{m}$ except the ones that encode the input $x$ and error $e$. We maintain a buffer of VSML RNN states from which we sample a batch during LA cloning and append one of the new states to the buffer. This ensures diversity across possible VSML RNN states during LA cloning.

**Batching for VSML RNNs**   In Section 3.2 we optimize a VSML RNN to implement backpropagation. To stabilize learning at meta test time we run the RNN on multiple data points (batch size 64) and then average their states corresponding to $w$ and $b$ as an analogue to batching in standard gradient descent.

**Stability during meta testing**   To prevent exploding states during meta testing we also clip the LSTM state between $-4$ and $4$.

**Bounded states in LSTMs**   In LSTMs the hidden state is bounded between $(-1, 1)$. For learning algorithm cloning, we'd like to support weights and biases beyond this range. This can be circumvented by choosing a constant, here $4$, by which we scale $w$ and $b$ down to store them in the context state. This is only relevant during learning algorithm cloning.

### C.2  Meta learning from scratch

**Hyperparameter search strategy**   The VSML hyper-parameters were searched using wandb's [5] Bayesian search during development. Parameters that lead to stable meta learning on MNIST were chosen. The final parameters were not further tuned and doing so may lead to additional performance gains. For the Meta RNN we picked parameters that matched VSML RNN as much as possible. For our SGD and SGD with Adam baselines, we performed a grid search over the learning rate on MNIST to find the best learning rates.

**Meta Training**   Meta training is done across 128 GPUs using ES as proposed by OpenAI [38] for a total of 10k steps. We use a population size of 1024, each population member is evaluated on one trajectory of 500 online examples. We use noise with a fixed standard deviation of $0.05$. To apply the estimated gradient, we use Adam with a learning rate of $0.025$ and betas set to $0.9$ and $0.999$. We have run similar experiments (where GPU memory is sufficient) with distributed gradient descent on 8 GPUs which led to less stable training but qualitatively similar results with appropriate early stopping and gradient clipping.

**VSML RNN architecture**    Each sub-RNN has a state size of $N = 16$ and messages are sized $N' = N'' = 8$. We only use a single layer between the input and prediction, thus $A$ equals the flattened input image dimension and $B = 10$ for the predicted logits. The outputs are squashed between $\pm 100$ using tanh. We run this layer two ticks per input. States are initialized randomly from independent standard normals.

**SGD baseline architecture and learning rate**    The deep SGD baseline uses a hidden layer of size 160, resulting in approximately $125k$ parameters on MNIST to match the number of state dimensions of the VSML RNN. We use a tanh activation function to match the LSTM setup. The tuned learning rate used for vanilla SGD is $10^{-2}$ and $10^{-3}$ for Adam.

**Meta RNN baseline**    We use an LSTM hidden size of 16 and an input size of $|\text{image}| + |\text{error}|$ where $|\text{error}|$ corresponds to the output size. Inputs are padded to be equal size across all meta training datasets. This results in about $100k$ to $150k$ parameters.

**Hebbian fast weight baseline**    We compare to a Hebbian fast weight baseline as described in Miconi et al. [25] where a single layer is adapted using learned synaptic plasticity. A single layer is adapted using Oja's rule by feeding the prediction errors and label as additional inputs.

**Specialization through RNN coordinates**    In addition to the recurrent inputs and inputs from the interaction term, each sub-RNN can be fed its coordinates $a, b$, position in time, or position in the layer stack. This may allow for (1) specialization, akin to the specialization of biological neurons, and (2) for implicitly meta learning neural architectures by suppressing outputs of sub-RNNs based on positional information. In our experiments, we have not yet observed any benefits of this approach and leave this to future work.

**Meta learning batched LAs**    In our meta learning from scratch experiments, we discovered online learning algorithms (similar to Meta RNNs [16, 56, 10]). We demonstrated high sample efficiency but the final performance trails the one of batched SGD training. In future experiments, we also want to investigate a batched variant. Every tick we could average a subset of each state $s_{ab}$ across multiple parallel running VSML RNNs. This would allow for meta learning batched LAs from scratch.

**Optimizing final prediction error vs sum of all errors**    In our experiments we are interested in sample efficient learning, i.e., the model making good predictions as early as possible in training. This is encouraged by minimizing the sum of all prediction errors throughout training. If only good final performance is desired, optimizing solely final prediction error or a weighting of prediction errors is an interesting alternative to be investigated in the future.

**Recursive replacement of weights**    Variable sharing in NNs by replacing each weight with an LSTM introduces new meta variables $V_M$. Those variables themselves may be replaced again by LSTMs, yielding a multi-level hierarchy with arbitrary depth. We leave the exploration of such hierarchies to future work.

**Alternative sparse shared weight matrices**    In this paper, we have focused on a version of VSML where the sparse shared weight matrix is defined by many RNNs that pass messages. Alternative ways of structuring variable sharing and sparsity may lead to different kinds of learning algorithms. Investigating these alternatives or even searching the space of variable sharing and sparsity patterns are interesting directions for future research.

**Meta Testing algorithm**    Meta testing corresponds to unrolling the VSML RNNs. The learning algorithm is encoded purely in the recurrent dynamics. See Algorithm 2 for pseudo-code.

**Algorithm 2** VSML: Meta Testing

---

**Require:** Dataset $D = \{(x_i, y_i)\}$, LSTM parameters $V_M$

$\quad V_L = \{s_{ab}^{(k)}\} \leftarrow$ initialize LSTM states $\quad \forall a, b, k$

$\quad$ **for** $(x, y) \in \{(x_1, y_1), \ldots, (x_T, y_T)\} \subset D$ **do** $\qquad\qquad\qquad\qquad$ ▷ Inner loop over $T$ examples

$\qquad \overrightarrow{m}_{a1}^{(1)} := x_a \quad \forall a$ $\qquad\qquad\qquad\qquad\qquad\qquad\qquad$ ▷ Initialize from input image x

$\qquad$ **for** $k \in \{1, \ldots, K\}$ **do** $\qquad\qquad\qquad\qquad\qquad\qquad\qquad\qquad$ ▷ Iterating over $K$ layers

$\qquad\qquad s_{ab}^{(k)} \leftarrow f_{RNN}(s_{ab}^{(k)}, \overrightarrow{m}_a^{(k)}, \overleftarrow{m}_b^{(k)}) \quad \forall a, b$ $\qquad\qquad\qquad\qquad$ ▷ Equation 7

$\qquad\qquad \overrightarrow{m}_b^{(k+1)} := \sum_{a'} f_{\overrightarrow{m}}(s_{a'b}^{(k)}) \quad \forall b$ $\qquad\qquad\qquad\qquad$ ▷ Create forward message

$\qquad\qquad \overleftarrow{m}_a^{(k-1)} := \sum_{b'} f_{\overleftarrow{m}}(s_{ab'}^{(k)}) \quad \forall a$ $\qquad\qquad\qquad\qquad$ ▷ Create backward message

$\qquad \hat{y}_a := \overrightarrow{m}_{a1}^{(K+1)} \quad \forall a$ $\qquad\qquad\qquad\qquad\qquad\qquad\qquad$ ▷ Read output

$\qquad e := \nabla_{\hat{y}} L(\hat{y}, y)$ $\qquad\qquad\qquad\qquad$ ▷ Compute error at outputs using loss $L$

$\qquad \overleftarrow{m}_{b1}^{(K)} := e_b \quad \forall b$ $\qquad\qquad\qquad\qquad\qquad\qquad\qquad$ ▷ Input errors

---

# D Other relationships to previous work

## D.1 VSML as distributed memory

Compared to other works with additional external memory mechanisms [53, 29, 40, 27, 42], VSML can also be viewed as having memory distributed across the network. The memory writing and reading mechanism implemented in the meta variables $V_M$ is shared across the network.

## D.2 Connection to modular learning

Our sub-LSTMs can also be framed as *modules* that have some shared meta variables $V_M$ and distinct learned variables $V_L$. Previous works in modular learning [49, 37, 19] were motivated by learning experts with unique parameters that are conditionally selected to suit the current task or context. In contrast, VSML has recurrent modules that share the same parameters $V_M$ to resemble a learning algorithm. There is no explicit conditional selection of modules, although it could emerge based on activations or be facilitated via additional attention mechanisms.

## D.3 Connection to self-organization and complex systems

In self-organizing systems, global behavior emerges from the behavior of many local systems such as cellular automata [8] and their recent neural variants [28, 52]. VSML can be seen as such a self-organizing system where many sub-RNNs induce the emergence of a global learning algorithm.