# OpenReview forum: "Meta Learning Backpropagation And Improving It"
_NeurIPS.cc/2021/Conference — NeurIPS 2021 Poster_

### Official Review · Reviewer_oKHU · 2021-07-08

**Rating:** 7
**Confidence:** 3

**Summary:**

This paper proposes VSML RNN, an architecture for RNNs that is suitable for meta-learning problems in which the RNN activations represent task-specific quantities and RNN weights represent task-independent quantities. The authors use the insight that RNNs for meta learning (meta RNNs as called in the paper) perform better when the meta-parameters are constrained relative to the task-specific parameters. VSML RNN is essentially an RNN with a particular weight sharing and sparsity structure. The authors demonstrate that VSML RNN can implement *within its hidden state* backpropagation on a smaller model, where the parameters of the smaller model are represented in the VSML RNN's hidden state. Empirically, the authors show that the learning algorithm implemented by a VSML RNN can outperform baselines, and can learn faster than gradient descent. The authors also demonstrate that the learning algorithm is transferable to unseen problems.

**Limitations And Societal Impact:**

The limitations and impact of this work are well-discussed in section 7.

**Main Review:**

**Originality**
The proposed approach appears novel. Unlike prior work on meta RNNs, the authors recognize that overparameterization of RNNs can lead to learning algorithms implemented by the RNN to overfit to the problems on which it is trained. The proposed method is explicitly designed to avoid such overfitting. Unlike prior work on learning learning rules, VSML RNN does not impose a particular architecture on  the task-specific learning algorithm.

**Quality**
The theoretical claims in the paper appear well supported. However, the experimental results are limited, particularly when it comes to comparisons with other meta RNN approaches. The paper would be strengthened by comparing VSML RNN to meta RNNs other than a standard meta RNN- for example, adding memory mechanisms to the RNN. Moreover, interpreting the learning algorithm implemented by VSML RNN would be helpful. The current analysis views the output of the model over the course of training, but finding a way to interpret the internal structure of VSML RNN activations would provide insight into the nature of the learned learning algorithm.

Furthermore, it is stated in the introduction that the results may be relevant in the field of biologically plausible learning algorithms. It would help if the authors could state how VSML RNN could be relevant here. Specifically, (how) can VSML RNN be used to discover biologically plausible learning rules in a way that existing approaches cannot? It seems that VSML RNN may be less useful for this compared to other approaches because it imposes fewer constraints on the form of the learning algorithm implemented by the RNN.


**Clarity**
VSML RNN is a difficult concept to understand- this makes clarity of the paper important. Unfortunately, there are some areas of the paper that are unclear or could be better explained. Specifically:
1) The authors state in section 3 that "A simple conceptual way of understanding VSML is taking a standard recurrent or feed-forward architecture and replacing all weights with small RNNs." This is hard to interpret since a weight is a parameter and a small RNN is a function- they do not seem comparable.
2) The distinction between the view of VSML RNN as an RNN (where s is a hidden state) and the view of it as implementing a learning algorithm on a neural network (where s could be a parameter) is not very clear. This is partly because section 3 tries to introduce both viewpoints: the equations look like typical RNN equations and suggest the first view, while Figure 1 encourages the second view. It may be helpful to structure the section to introduce the two viewpoints separately and then explain the mapping between them.
3) Visually illustrating the indexing of s would be helpful. Initially, it is not clear why the indexing in Equation 3 for example is constructed the way it is.
4) Similarly, visually illustrating how stacking VSML RNNs works would be helpful. Figure 1 is not very clear- the distinctions and equivalences between VSML RNN states, parameters, messages and graph edges is not visually apparent.


**Significance**
If the experimental results can be improved, VSML RNNs may become the state-of-the-art approach in the field of meta RNNs. The learning algorithms produced by VSML RNNs may also provide more insight into how to develop more general, better performing learning algorithms. Finally, VSML RNNs further unify meta RNNs and learning algorithms on neural networks, which may lead to more flexible meta-learning procedures.

**Time Spent Reviewing:**

2

---

> ### Author Response · Authors · 2021-08-10
> **Addressing baselines, clarity, and interpretability**
>
> Thanks a lot for your review, suggestions, and questions.
>
> Generally, we had the impression you are positively impressed by the paper and we hope that based on our comments below you are willing to reconsider your score.
>
> From our understanding, your main concerns are missing baselines, clarity, and interpretability. In the following, we attempt to address all of these issues.
>
> Also, note that we have extended VSML to CNN architectures and CIFAR10 based on the feedback from the other reviewers.
>
> ### External memory baseline
>
> > The paper would be strengthened by comparing VSML RNN to meta RNNs other than a standard meta RNN- for example, adding memory mechanisms to the RNN.
>
> Based on your suggestion, we now include an additional Meta RNN baseline with external memory. We have implemented a recent version using external fast weight memory [1] that has proven successful.
>
> In this new figure ([https://ibb.co/16vf6Qb](https://ibb.co/16vf6Qb)) we have plotted the meta-test running accuracy for VSML and the external memory baseline on MNIST (within distribution) and Fashion MNIST (out of distribution) after meta training on either only MNIST or the set of MNIST, EMNIST, KMNIST, and a random classification dataset.
>
> We observe that when meta testing on MNIST, our baseline's performance is strong for both meta-training distributions. In contrast, generalization to Fashion MNIST is far behind VSML. This suggests similar overfitting behavior that we observed with standard MetaRNNs. On the other hand, with an increase in the size of the meta training distribution, the generalization performance of the external memory baseline increases to some extent (better than random predictions).
>
> We will add these results to the paper and also test other environment combinations.
>
> [1] Schlag, Imanol, Tsendsuren Munkhdalai, and Jürgen Schmidhuber. "Learning Associative Inference Using Fast Weight Memory." *arXiv preprint arXiv:2011.07831* (2020).
>
> ### Clarity
>
> Based on your suggestions regarding clarity, we have done many improvements. In addition to figure 1, we will include a more detailed figure. The new figure can be found here [https://ibb.co/C5YmtRw](https://ibb.co/C5YmtRw). In the figure, all LSTMs share their parameters $V_M$.
>
> > Clarity 3) Visually illustrating the indexing of s would be helpful. Initially, it is not clear why the indexing in Equation 3 for example is constructed the way it is.
>
> In the new figure, we have visually indexed all the states and messages for this example of a small 2x2 network with two layers.
>
> Messages are defined as
>
> $\boldsymbol{\overrightarrow m}_a^{(k+1)} :=$ (line break due to latex bug)
>
> $\sum_c \overrightarrow m(s_{ca}^{(k)})$
>
> $\boldsymbol{\overleftarrow m}_b^{(k-1)} :=$ (line break due to latex bug)
>
> $\sum_c \overleftarrow m(s_{bc}^{(k)})$
>
> We also highlighted in the figure where the $\sum_c$​​​​​ in equation 3 comes from with a green circle. We have improved the introduction of equation 3.
>
> **Stacking multiple layers vs 'recurrent' architecture**
>
> > Clarity 4) Similarly, visually illustrating how stacking VSML RNNs works would be helpful. Figure 1 is not very clear- the distinctions and equivalences between VSML RNN states, parameters, messages and graph edges is not visually apparent.
>
> The new figure should now also clarify the distinction between LSTM parameters, states, and messages.
>
> In standard RNNs, if we would use different weights at each tick, we'd have a multi-layer architecture. Similarly, in VSML, if we use different states $s_{ab}^{(k)}$​​​ for each layer $k$​​​, we obtain multiple layers. Furthermore, we can then vary the input dimensionality $A^{(k)}$​​​​ and output dimensionality $B^{(k)}$​​​​​​ for each layer. Conversely, if we share the same states across 'layers' we have a 'recurrent' architecture. Note that this 'recurrence' is in addition to the recurrence that already exists in each sub-RNN with states $s_{ab}^{(k)}$​.
>
> **Pseudo-code for Meta Training and Meta Testing**
>
> We will also include pseudo-code for meta training and meta testing in the camera-ready version:
>
> Meta Training
>
> - $V_M \leftarrow \textrm{initialize}$ (LSTM parameters)
> - Outer loop
>   - $s_{ab}^{(k)} \leftarrow \textrm{initialize}$ (Learned variables $V_L := \{s_{ab}^{(k)}\}$)
>   - Inner loop $t \in \{1, \ldots, T\}$ iterating over $T$ examples
>     - $\boldsymbol{\overrightarrow m}^{(1)}_{a0} := x_a$ (Initialize from input image x)
>     - for $k \in \{1, \ldots, K\}$ iterating over $K$ layers
>       - $s_{ab}^{(k)} \leftarrow f_{V_M}(s_{ab}^{(k)}, \boldsymbol{\overrightarrow m}_a^{(k)}, \boldsymbol{\overleftarrow m}_b^{(k)})$ (Equation 8)
>       - $\boldsymbol{\overrightarrow m}_a^{(k+1)} :=$ (Create forward message; line break due to latex bug)
>         - $\sum_c \overrightarrow m(s_{ca}^{(k)})$
>       - $\boldsymbol{\overleftarrow m}_b^{(k-1)} :=$ (Create backward message; line break due to latex bug)
>         - $\sum_c \overleftarrow m(s_{bc}^{(k)})$
>     - $y_a := \boldsymbol{\overrightarrow m}_{a0}^{(K+1)}$ (Read output)
>     - $e := \nabla_y L(y, \hat y)$ (Compute error at outputs using loss $L$)
>     - $\boldsymbol{\overleftarrow m}_{b0}^{(K)} := e_b$ (Input errors)
>   - $V_M \leftarrow V_M - \alpha \nabla_{V_M} \sum_{t=1}^{T} L(y(t), \hat y(t))$, obtaining $\nabla_{V_M}$ either by
>     - back-propagation through the inner loop OR
>     - evolution strategies, using a search distribution $p(V_M)$​
>
> Meta Testing (same as meta training, but only inner loop)
>
> - Given LSTM parameters $V_M$
> - $s_{ab}^{(k)} \leftarrow \textrm{initialize}$ (Learned variables $V_L := \{s_{ab}^{(k)}\}$)
> - Inner loop $t \in \{1, \ldots, T\}$ iterating over $T$ examples
>   - $\boldsymbol{\overrightarrow m}^{(1)}_{a0} := x_a$ (Initialize from input image x)
>   - for $k \in \{1, \ldots, K\}$ iterating over $K$ layers
>     - $s_{ab}^{(k)} \leftarrow f_{V_M}(s_{ab}^{(k)}, \boldsymbol{\overrightarrow m}_a^{(k)}, \boldsymbol{\overleftarrow m}_b^{(k)})$ (Equation 8)
>     - $\boldsymbol{\overrightarrow m}_a^{(k+1)} :=$ (Create forward message; line break due to latex bug)
>       - $\sum_c \overrightarrow m(s_{ca}^{(k)})$
>     - $\boldsymbol{\overleftarrow m}_b^{(k-1)} :=$ (Create backward message; line break due to latex bug)
>       - $\sum_c \overleftarrow m(s_{bc}^{(k)})$
>   - $y_a := \boldsymbol{\overrightarrow m}_{a0}^{(K+1)}$ (Read output)
>   - $e := \nabla_y L(y, \hat y)$ (Compute error at outputs using loss $L$)
>   - $\boldsymbol{\overleftarrow m}_{b0}^{(K)} := e_b$ (Input errors)
>
>
> > Clarity 1) The authors state in section 3 that "A simple conceptual way of understanding VSML is taking a standard recurrent or feed-forward architecture and replacing all weights with small RNNs." This is hard to interpret since a weight is a parameter and a small RNN is a function- they do not seem comparable.
>
> You are right, we should be more precise here. We replace the activation-weight multiplication operation with an RNN.
> From a computational graph perspective: In a NN, for each weight, there is an input activation, a multiplication operation, and an output. We replace this with an RNN that receives a message (which contains something akin to an input activation), does some computation based on its state, and outputs a new message. Thanks for the pointer, we clarified this in the paper.
>
> >  Clarity 2) The distinction between the view of VSML RNN as an RNN (where s is a hidden state) and the view of it as implementing a learning algorithm on a neural network (where s could be a parameter) is not very clear. [...] It may be helpful to structure the section to introduce the two viewpoints separately and then explain the mapping between them.
>
> This is a great suggestion and we will make sure to restructure this section accordingly.
>
> ### Interpretability
>
> > Moreover, interpreting the learning algorithm implemented by VSML RNN would be helpful. The current analysis views the output of the model over the course of training, but finding a way to interpret the internal structure of VSML RNN activations would provide insight into the nature of the learned learning algorithm.
>
> We agree that it would be very interesting to work on interpretability for VSML. In section 5 we have already attempted to look at the behavior by analyzing model outputs. Beyond this, one could for example investigate how the states change as a response to prediction error. That said, we think it will be hard to arrive at a holistic understanding and it involves many open research questions. It is therefore beyond the scope of the current work but could be an interesting future direction.
>
> ### Biological plausibility
>
> > it is stated in the introduction that the results may be relevant in the field of biologically plausible learning algorithms. [...] Specifically, (how) can VSML RNN be used to discover biologically plausible learning rules in a way that existing approaches cannot? It seems that VSML RNN may be less useful for this compared to other approaches because it imposes fewer constraints on the form of the learning algorithm implemented by the RNN.
>
> We did not focus on interpretability and biological constraints. Thus, we agree that biologically plausible learning is less feasible to assess. We want to emphasize that it is not a contribution of our paper and we will revise our statement accordingly.
>
> We would like to elaborate on the interpretation that we had in mind: From the perspective that the brain may learn using different mechanisms than backpropagation, our paper presents the possibility of automatically discovering new learning algorithms alternative to backprop. While there is existing work on meta-learning synaptic update rules, different to previous work, our paper focuses on meta-learning learning algorithms that are more general-purpose and can be reused across datasets. Furthermore, we demonstrated how backprop can be embedded in the recurrent dynamics of an RNN.
>
> Thanks again, and please let us know should you have any other concerns.

---

> > ### Comment · Reviewer_oKHU · 2021-08-28
> > **Thank you for your detailed response!**
> >
> > I find the additional experiments and improvements to the clarity of the work quite valuable, and I have significantly raised my rating accordingly.

---

### Official Review · Reviewer_MHDx · 2021-07-10

**Rating:** 6
**Confidence:** 3

**Summary:**

This paper proposes variable shared meta learning,  and implements the network forward and backward computation with RNN.  The proposed method can also be used for learning general learning algorithms. Experiments on MNIST and Fashion MNIST dataset demonstrate the effectiveness of the proposed method.


**Limitations And Societal Impact:**

The authors sufficiently discuss the limitations and societal impact of the work.

**Main Review:**

Originality:

The proposed method is novel compared to existing works. The authors clearly describe the difference w.r.t to previous contributions and well-positioned with related works.

Quality:

This submission is technically sound.  The main claims are supported by the experimental results. The methods are carefully designed and demonstrated its advantage.

Clarity:

This paper is clearly written and well-structured.

Significance:

1. The experimental section only use simple datasets, e.g., MNIST. I would like to see more real dataset, e.g., CIFAR10 or CIFAR100. This would improve the applicability on the real tasks.

2.  RNN is hard to train in practice, how to ensure the stability of the proposed methods in real datasets since the RNNs are interactived with each other in the proposed method.

3.  I wonder if the proposed method can be applied on other network architectures, e.g., convolutional network. This will broaden the applicability of the proposed method.

4. I would like to see what is the performance on larger-scale network structure, currently only experiment on small-scale networks.

**Time Spent Reviewing:**

7 hours

---

> ### Author Response · Authors · 2021-08-10
> **Addressing applicability to convolutions, more difficult datasets, larger-scale network architectures**
>
> Thanks a lot for your review, suggestions, and questions.
>
> From our understanding, your main concerns are applicability to convolutions, more difficult datasets like CIFAR10, and larger-scale network architectures. In the following, we attempt to address all of these issues.
>
> > 3 I wonder if the proposed method can be applied on other network architectures, e.g., convolutional network. This will broaden the applicability of the proposed method.
>
> Indeed, our method can also be used for convolutional layers. To demonstrate this, we have now also implemented a CNN version of VSML with up to three convolutions. This is done by replacing each weight in the kernel with a multi-dimensional RNN state and replacing the kernel multiplications with VSML sub-RNNs. On our existing datasets, it performs similar to the fully connected architecture, as can be seen in this new figure [https://ibb.co/kytB5xy](https://ibb.co/kytB5xy).
>
> > 1 The experimental section only use simple datasets, e.g., MNIST. I would like to see more real dataset, e.g., CIFAR10 or CIFAR100. This would improve the applicability on the real tasks.
>
> Based on your comment, we also applied our CNN variant to CIFAR10. Note that in our paper we were interested in the online learning setting (similar to Meta RNNs). This is a challenging problem on which gradient descent with back-propagation also struggles. Many consecutive examples ($> 10^{5}$​​​ steps) are required for learning. Online performance is generally lower than in the batched setting which we do not explore here.
>
> When meta training on CIFAR10 (see new figure [https://ibb.co/qYhT2zT](https://ibb.co/qYhT2zT)) we observe that meta-test-time learning on CIFAR is initially faster compared to SGD while still  generalizing to Fashion MNIST. On the other hand, with a sufficiently large meta training distribution, we would hope to see similar generalization to CIFAR10 when CIFAR10 is unseen. As visible in both plots, learning speed decreases at some point. One challenge is that during meta training we only unroll the inner loop for 500 examples (dashed line), not explicitly optimizing learning beyond that point. This could be addressed in future work by starting the inner loop from previous states that already have some learning progress (similar to truncated backpropagation).
>
> > 2 RNN is hard to train in practice, how to ensure the stability of the proposed methods in real datasets since the RNNs are interactived with each other in the proposed method.
>
> This is a great comment. First of all, we are using LSTMs in practice to stabilize the recurrent dynamics. Furthermore, we observed that the meta-loss-landscapes can be difficult to optimize. This is why we opted for evolution strategies during meta learning which produces more stable meta updates. Also, see [1] for similar observations. Generally, future investigations into meta-loss-landscapes generated by VSML will be valuable.
>
> [1] Metz, L., Maheswaranathan, N., Nixon, J., Freeman, C. D., & Sohl-Dickstein, J. (2018). *Understanding and correcting pathologies in the training of learned optimizers*. http://arxiv.org/abs/1810.10180
>
> > 4 I would like to see what is the performance on larger-scale network structure, currently only experiment on small-scale networks.
>
> Based on your comment, we have increased the complexity of our architecture by adding convolutional layers. We acknowledge that larger-scale network structures would be interesting but those would also result in larger compute requirements and training times.
>
> Thanks again, and please let us know should you have any other concerns.

---

### Official Review · Reviewer_v8jj · 2021-07-14

**Rating:** 8
**Confidence:** 4

**Summary:**

The authors in this paper propose to learn learning algorithms (LAs) like backpropagation using an RNN/LSTM to replace all the weights of a network. Subsequently, they use LA on a different dataset to show that the LA has generalized.

**Limitations And Societal Impact:**

The paper is hard to read in many places. For, e.g., in eq-3, the sudden use of $s_{cai}$ without defining the term kept me guessing. Also, it is unclear how the error passing is learned using gradient descent for the backward pass emulating backpropagation.

I would recommend adding a step-by-step algorithm to the paper to make things more straightforward. Including the gradient descent update for both forward and backward weights would be helpful.
Also, given the paper's claim of learning the meta-algorithm, I think the authors should also include the results of using this method on conv networks.

A single ablation using a small conv network trained on Cifar using LA learned over FC layers would significantly improve the impact of the paper.

**Main Review:**

The authors model each weight with an LSTM's state. The forward and backward passes of the NN are coded into the LSTM's state updates. The work is novel and is an attempt to find algorithms that can replace backpropagation for training NN. The authors show that the learned backpropagation can learn on other datasets. They also show that the learning is faster and can learn with lesser examples.

**Time Spent Reviewing:**

6

---

> ### Author Response · Authors · 2021-08-10
> **Addressing clarity and convolutions**
>
> Thanks a lot for your review, suggestions, and questions.
>
> From our understanding, your main concerns are about the clarity of the paper and whether the method can be extended to convolutions. In the following, we attempt to address both issues.
>
> > I would recommend adding a step-by-step algorithm to the paper to make things more straightforward.
>
> Regarding clarity, we have made several changes following your suggestions. Below are the pseudo-code algorithms for meta training and meta testing and we will include those in the paper.
>
> **Meta Training**
>
> - $V_M \leftarrow \textrm{initialize}$ (LSTM parameters)
> - Outer loop
>   - $s_{ab}^{(k)} \leftarrow \textrm{initialize}$ (Learned variables $V_L := \{s_{ab}^{(k)}\}$)
>   - Inner loop $t \in \{1, \ldots, T\}$ iterating over $T$ examples
>     - $\boldsymbol{\overrightarrow m}^{(1)}_{a0} := x_a$ (Initialize from input image x)
>     - for $k \in \{1, \ldots, K\}$ iterating over $K$ layers
>       - $s_{ab}^{(k)} \leftarrow f_{V_M}(s_{ab}^{(k)}, \boldsymbol{\overrightarrow m}_a^{(k)}, \boldsymbol{\overleftarrow m}_b^{(k)})$ (Equation 8)
>       - $\boldsymbol{\overrightarrow m}_a^{(k+1)} :=$ (Create forward message; line break due to latex bug)
>         - $\sum_c \overrightarrow m(s_{ca}^{(k)})$
>       - $\boldsymbol{\overleftarrow m}_b^{(k-1)} :=$ (Create backward message; line break due to latex bug)
>         - $\sum_c \overleftarrow m(s_{bc}^{(k)})$
>     - $y_a := \boldsymbol{\overrightarrow m}_{a0}^{(K+1)}$ (Read output)
>     - $e := \nabla_y L(y, \hat y)$ (Compute error at outputs using loss $L$)
>     - $\boldsymbol{\overleftarrow m}_{b0}^{(K)} := e_b$ (Input errors)
>   - $V_M \leftarrow V_M - \alpha \nabla_{V_M} \sum_{t=1}^{T} L(y(t), \hat y(t))$, obtaining $\nabla_{V_M}$ either by
>     - back-propagation through the inner loop OR
>     - evolution strategies, using a search distribution $p(V_M)$​
>
> **Meta Testing (same as meta training, but only inner loop)**
>
> - Given LSTM parameters $V_M$
> - $s_{ab}^{(k)} \leftarrow \textrm{initialize}$ (Learned variables $V_L := \{s_{ab}^{(k)}\}$)
> - Inner loop $t \in \{1, \ldots, T\}$ iterating over $T$ examples
>   - $\boldsymbol{\overrightarrow m}^{(1)}_{a0} := x_a$ (Initialize from input image x)
>   - for $k \in \{1, \ldots, K\}$ iterating over $K$ layers
>     - $s_{ab}^{(k)} \leftarrow f_{V_M}(s_{ab}^{(k)}, \boldsymbol{\overrightarrow m}_a^{(k)}, \boldsymbol{\overleftarrow m}_b^{(k)})$ (Equation 8)
>     - $\boldsymbol{\overrightarrow m}_a^{(k+1)} :=$ (Create forward message; line break due to latex bug)
>       - $\sum_c \overrightarrow m(s_{ca}^{(k)})$
>     - $\boldsymbol{\overleftarrow m}_b^{(k-1)} :=$ (Create backward message; line break due to latex bug)
>       - $\sum_c \overleftarrow m(s_{bc}^{(k)})$
>   - $y_a := \boldsymbol{\overrightarrow m}_{a0}^{(K+1)}$ (Read output)
>   - $e := \nabla_y L(y, \hat y)$ (Compute error at outputs using loss $L$)
>   - $\boldsymbol{\overleftarrow m}_{b0}^{(K)} := e_b$ (Input errors)
>
>
> > The paper is hard to read in many places. For, e.g., in eq-3, the sudden use of scai without defining the term kept me guessing.
>
> We agree that we should have been more verbose about equation 3. We now have adapted the introduction of equation 3 and include a new figure that visualizes the term. The new figure can be found here [https://ibb.co/C5YmtRw](https://ibb.co/C5YmtRw).
>
> In the figure, all LSTMs share their parameters $V_M$​​​​​​. We have also visually indexed all the states and messages for this example of a small 2x2 network with two layers.
>
> Messages are defined as
>
> $\boldsymbol{\overrightarrow m}_a^{(k+1)} :=$ (line break due to latex bug)
>
> $\sum_c \overrightarrow m(s_{ca}^{(k)})$
>
> $\boldsymbol{\overleftarrow m}_b^{(k-1)} :=$ (line break due to latex bug)
>
> $\sum_c \overleftarrow m(s_{bc}^{(k)})$
>
> We also used green circles in the figure to highlight where the $\sum_c$​​​​​​ in equation 3 originates.
>
> Equation 3 for reference:
> $$s_{abj} \leftarrow \sigma(\sum_i s_{abi} W_{ij} + \underbrace{\sum_{c,i} s_{cai} C_{ij}}_{\text{interactions}})$$
>
> The term $\sum_{c,i} s_{cai} C_{ij}$​​​ where $c \in \{1, \ldots, A\}$​​​ and $C \in \mathbb{R}^{N \times N}$​​​ is used to make the different sub-RNNs interact with each other like weights are connected in a standard fully connected layer. We can decompose it into $\sum_c \sum_i s_{cai}C_{ij}$
> where $\overrightarrow m (s_{ca}) = \sum_i s_{cai} C_{ij}$​​​ is a transformation of the state (with $\overrightarrow m: \mathbb{R}^N \to \mathbb{R}^N$​) to form the forward message which is then sum-reduced $\sum_c$​​​​(compare green circles in figure above).
>
> > Also, it is unclear how the error passing is learned using gradient descent for the backward pass emulating backpropagation.
>
> Each layer passes information forward ($\overrightarrow m$​​) and backward ($\overleftarrow m$​​​​) in the network. This allows learning to receive an input, produce forward activations, receive the error, and pass a new error signal backward (all defined by the LSTM parameters). This is depicted in figure 2. It is essentially the generalization of a computational graph that autograd would produce.
>
> >  Including the gradient descent update for both forward and backward weights would be helpful.
>
> We hope the pseudo-code above resolves this issue. There are only gradient descent updates for the meta-parameters $V_M$​​​​​​​, ie the parameters of the sub-RNN / sub-LSTM.
>
> > Also, given the paper's claim of learning the meta-algorithm, I think the authors should also include the results of using this method on conv networks.
>
> Based on your suggestion, we have now also implemented a CNN version of VSML with up to three convolutions. This is done by replacing each weight in the kernel with a multi-dimensional RNN state and replacing the kernel multiplications with VSML sub-RNNs. On our existing datasets, it performs similar to the fully connected architecture, as can be seen in this new figure [https://ibb.co/kytB5xy](https://ibb.co/kytB5xy).
>
> > A single ablation using a small conv network trained on Cifar using LA learned over FC layers would significantly improve the impact of the paper.
>
> Based on your other comment, we also applied our CNN variant to CIFAR10. Note that in our paper we were interested in the online learning setting (similar to Meta RNNs). This is a challenging problem on which gradient descent with back-propagation also struggles. Many consecutive examples ($> 10^{5}$​​​ steps) are required for learning. Online performance is generally lower than in the batched setting which we do not explore here.
>
> When meta training on CIFAR10 (see new figure [https://ibb.co/qYhT2zT](https://ibb.co/qYhT2zT)) we observe that meta-test-time learning on CIFAR is initially faster compared to SGD while still  generalizing to Fashion MNIST. On the other hand, with a sufficiently large meta training distribution, we would hope to see similar generalization to CIFAR10 when CIFAR10 is unseen. As visible in both plots, learning speed decreases at some point. One challenge is that during meta training we only unroll the inner loop for 500 examples (dashed line), not explicitly optimizing learning beyond that point. This could be addressed in future work by starting the inner loop from previous states that already have some learning progress (similar to truncated backpropagation).
>
> Thanks again, and please let us know should you have any other concerns.

---

> > ### Comment · Reviewer_v8jj · 2021-09-10
> > **Scores updated**
> >
> > I still have a few concerns but as a reviewer, I feel the need to support bold research even if it has small gaps.

---

### Official Review · Reviewer_9hFV · 2021-07-16

**Rating:** 6
**Confidence:** 3

**Summary:**

The paper proposes to meta-learn the backpropagations by representing weights of a network by small RNNs. Different RNNs share the weights, and the weights can be thought of as representing the states of RNN. The approach can thus implement back-propagation solely in the forward dynamics of an RNN.

**Limitations And Societal Impact:**

No, the authors haven't addressed it. Though I don't feel that the paper would have any negative societal impact.

**Main Review:**

I like the general idea of the paper but have a few concerns regarding the approach and its motivation -

1. The paper uses evolutionary algorithms to train their model. But evolutionary algorithms can also be used to directly learn the weights of a neural network, and it has been shown to achieve nice results.[1] What's the advantage of using VSML as an alternative to back-propagation?

2. For meta-learning can the authors show results over Omniglot dataset, which is considered a more standard dataset for meta-learning.

3. Since VSML is computationally expensive, what's the advantage of using it instead of back-propagation?

[1] Designing neural networks through neuroevolution. Stanley et al.

**Time Spent Reviewing:**

3

---

> ### Author Response · Authors · 2021-08-10
> **Addressing motivation and choice of benchmark**
>
> Thanks a lot for your review, suggestions, and questions.
>
> From our understanding, your main concerns are with the motivation of the paper and our choice of benchmark. In the following, we attempt to address both issues.
>
> > 1 The paper uses evolutionary algorithms to train their model. But evolutionary algorithms can also be used to directly learn the weights of a neural network, and it has been shown to achieve nice results.[1] What's the advantage of using VSML as an alternative to back-propagation?
>
> Our paper uses evolutionary algorithms to train the learning algorithm instead of the solution to a problem directly. Evolution is indeed *one possible* good learning algorithm (gradient descent with backpropagation another).
>
> One way of viewing our procedure is as amortization -- we incur a one-time high computational cost at meta training to later learn more cheaply at meta-test-time. This enables going beyond a fixed human-engineered learning algorithm. In the process, we show that more sample efficient learning algorithms can be discovered. Crucially, our paper focuses on generalization of the resulting learning algorithm, much more than related work. If generalization is limited we need to run high-cost meta-training more frequently.
>
> > 3 Since VSML is computationally expensive, what's the advantage of using it instead of back-propagation?
>
> We would like to highlight again that while meta-training is very expensive, meta-testing is considerably cheaper. Our VSML RNN's complexity is only scaled by a constant $N^2$​​​​​​​ compared to a standard neural network. The complexity is in $O(WN^2)$​​​​​​​​​ where $W$​​​​​​​ are the standard number of parameters in a neural network and $N$​​​​​​​​ is the hidden size of the small RNN each weight gets replaced with ($N = 16$​​​​​​​​​​​​​ in our experiments). Our contributions are two-fold: Firstly, we show that we can go beyond back-propagation as a fixed routine. Instead, powerful and generalizing learning algorithms can be discovered via meta learning. Furthermore, we show that our meta-learned learning algorithm can yield improvements in sample efficiency over back-propagation.
>
> > 2 For meta-learning can the authors show results over Omniglot dataset, which is considered a more standard dataset for meta-learning.
>
> When choosing our benchmark, our thought process was to pick datasets that are particularly challenging in terms of generalization. Our meta learning framework should allow meta-training on e.g. MNIST and produce a learning algorithm that generalizes to Fashion MNIST or even a randomly generated classification dataset (also KMNIST, EMNIST, our Sign-Dataset, etc). This is different from the Omniglot dataset where all images are from varying alphabets and there is a large meta training distribution that covers the meta test set well.
>
> > Limitations And Societal Impact: No, the authors haven't addressed it. Though I don't feel that the paper would have any negative societal impact.
>
> We tried to address societal impact in a short note in the paper checklist at the end of the PDF.
>
> Thanks again, and please let us know should you have any other concerns.

---

> > ### Comment · Reviewer_9hFV · 2021-08-18
> > **Response to the authors**
> >
> > I thank the authors for trying to address my concerns.
> >
> > 1. I still have my reservations over the advantage provided by the method over directly using evolutionary algorithms.  By directly using evolutionary algorithms over the weights, we are again not really using a fixed human-engineered algorithm, same as the current paper motivation. But in addition evolutionary algorithms lead to good empirical performance even over large-scale datasets.
> >
> > 2. I am not convinced by the experimental choice of not using the Omniglot dataset for meta-learning experiments. I agree that the experimental choice made by you is a nice one, but still results should have been shown over the Omniglot dataset as it's a standard dataset.

---

> > > ### Author Response · Authors · 2021-08-19
> > > **Meta learning with evolutionary algorithms and Omniglot experiments**
> > >
> > > Thank you for your response and the continuation of the discussion.
> > >
> > > > I still have my reservations over the advantage provided by the method over directly using evolutionary algorithms. By directly using evolutionary algorithms over the weights, we are again not really using a fixed human-engineered algorithm, same as the current paper motivation. But in addition evolutionary algorithms lead to good empirical performance even over large-scale datasets.
> > >
> > > Perhaps there is a misunderstanding about what you mean by 'directly using evolutionary algorithms over the weights'. If we used an evolutionary algorithm such as ES [1] to directly update the weights of a neural network (e.g. a feed-forward classifier), we would use a specific algorithm (ES) to update the weights that is fixed throughout training. The way the weights are updated (ES) is human-engineered. In comparison, our paper uses ES for _meta-learning_, i.e. to learn a learning algorithm that defines how 'weights' (RNN states in our case) are updated when digesting datasets such as MNIST. By meta-learning in this way, we are able to improve over existing learning algorithms, such as SGD. We would appreciate it if you could elaborate on your concerns.
> > >
> > > [1] Salimans, T., Ho, J., Chen, X., Sidor, S., & Sutskever, I. (2017). Evolution Strategies as a Scalable Alternative to Reinforcement Learning.
> > >
> > > > I am not convinced by the experimental choice of not using the Omniglot dataset for meta-learning experiments. I agree that the experimental choice made by you is a nice one, but still results should have been shown over the Omniglot dataset as it's a standard dataset.
> > >
> > > Since you remain concerned, we have now also included Omniglot results to our current benchmarks in the paper. You can find the new figure here [https://ibb.co/P5HMtFd](https://ibb.co/P5HMtFd).
> > > On Omniglot, our experimental setting corresponds to the common 5-way, 1-shot setting [6]: In each episode, we select 5 random classes and sample 1 instance each and show it with the label and prediction error to the network. Then, we sample a new random test instance from one of the 5 classes and meta-train to minimize the cross-entropy on that example. At meta-test time we use unseen alphabets (classes) from the test set and report the accuracy of the test instance across 100 episodes.
> > >
> > > The results ([https://ibb.co/P5HMtFd](https://ibb.co/P5HMtFd)) nicely demonstrate how common baselines such as the Meta RNN [2-4] or a Meta RNN with external memory [5] work well in an Omniglot setting, yet fail when the gap increases between meta-train and meta-test, thus requiring stronger generalization (paper figure 4, 6, and [https://ibb.co/16vf6Qb](https://ibb.co/16vf6Qb) for reviewer oKHU). In contrast, VSML generalizes well to unseen datasets, e.g. Fashion MNIST, although it does learn more slowly on Omniglot. We will discuss this trade-off in the paper. Finally, these new results demonstrate how VSML learns significantly faster on Omniglot compared to SGD with Adam, thus highlighting the benefits of the meta-learning approach adopted in this work.
> > >
> > > We hope these additional experiments on Omniglot remove any remaining doubt about the working of our method.
> > >
> > > [2] Hochreiter, S., Younger, A. S., & Conwell, P. R. (2001). Learning to learn using gradient descent. International Conference on Artificial Neural Networks.
> > >
> > > [3] Wang, J. X., Kurth-Nelson, Z., Tirumala, D., Soyer, H., Leibo, J. Z., Munos, R., Blundell, C., Kumaran, D., & Botvinick, M. (2016). Learning to reinforcement learn.
> > >
> > > [4] Duan, Y., Schulman, J., Chen, X., Bartlett, P. L., Sutskever, I., & Abbeel, P. (2016). RL^2: Fast Reinforcement Learning via Slow Reinforcement Learning.
> > >
> > > [5] Schlag, Imanol, Tsendsuren Munkhdalai, & Jürgen Schmidhuber. (2020). Learning Associative Inference Using Fast Weight Memory.
> > >
> > > [6] Miconi, T., Clune, J., & Stanley, K. O. (2018). Differentiable plasticity: training plastic neural networks with backpropagation. In International Conference on Machine Learning.

---

> > > > ### Comment · Reviewer_9hFV · 2021-08-29
> > > > **Thanks for the experiments**
> > > >
> > > > I thank the authors for the experiments. I have accordingly updated my ratings

---

### Author Response · Authors · 2021-08-27
**Any remaining concerns?**

Thanks again for all the feedback from the reviewers.
We hope our response was satisfactory.
Given that the discussion period is ending soon, we would like to ask the reviewers to raise any concerns they may still have at this point?

---

### Decision · Program_Chairs · 2021-09-27

**Decision:**

Accept (Poster)

**Comment:**

The reviewers were in agreement that the method developed were novel and well evaluated, particularly with the additional details and experimental results that came up in the discussion. These results should be incorporated in the final version of the paper. Overall this paper should be of significant interest to the NeurIPS community.